# Elucidating the genetic mechanisms governing cytosine base editing outcomes through CRISPRi screens

Sifeng Gu[1,4], Zsolt Bodai[1,4], Rachel A. Anderson [1], Hei Yu Annika So [1], Quinn T. Cowan [1] & Alexis C. Komor [1,2,3] ✉

Cytosine base editors enable programmable and efficient genome editing using an intermediate featuring a U•G mismatch across from a DNA nick. This intermediate facilitates two major outcomes, C•G to T•A and C•G to G•C point mutations, and it is not currently well-understood which DNA repair factors are involved. Here, we couple reporters for cytosine base editing activity with knockdown of 2015 DNA processing genes to identify genes involved in these two outcomes. Our data suggest that mismatch repair factors facilitate C•G to T•A outcomes, while C•G to G•C outcomes are mediated by RFWD3, an E3 ubiquitin ligase. We also propose that XPF, a 3'-flap endonuclease, and LIG3, a DNA ligase, are involved in repairing the intermediate back to the original C•G base pair. Our results demonstrate that competition and collaboration among different DNA repair pathways shape cytosine base editing outcomes.

Base editing technologies facilitate the efficient and precise introduction of point mutations into the genome without the use of DNA double-strand breaks (DSBs). Currently, there are two major classes of base editors, cytosine base editors (CBEs) and adenine base editors (ABEs)[1]. CBEs are generally protein fusion constructs consisting of a catalytically impaired Cas nuclease linked to a cytidine deaminase enzyme specific for single-stranded DNA (ssDNA) substrates. Upon localization of the Cas protein to its target DNA sequence, hybridization of a short piece of RNA called the single guide RNA (sgRNA) to its complimentary DNA strand exposes a small stretch of ssDNA on the non-complementary strand to the deaminase enzyme (Fig. 1a). This exposure allows the cytidine deaminase to gain access to and deaminate cytosine bases to uracil within this ssDNA window[2]. Furthermore, most CBEs use a Cas nickase (Cas9n), which introduces a single-strand break on the sgRNA-complementary stand. Consequently, CBEs generate a unique type of DNA damage, featuring a U•G mismatch with a DNA nick 5' upstream from the mismatch on the G-containing strand (Fig. 1a).

Cellular processing of this DNA damage intermediate leads to two major mutational outcomes, C•G to T•A and C•G to G•C conversions (Fig. 1a). C•G to A•T conversions and indel products can also be introduced, although at much lower rates. It has been previously shown that the endogenous DNA repair enzyme uracil-DNA glycosylase (UNG) plays an important role in the relative distribution of these outcomes by excising the uracil intermediate into an abasic site, which leads to C•G to non-T•A outcomes[3,4]. Therefore, canonical C•G to T•A base editors (which we will refer to as CBEs here) are covalently fused to one or more copies of an uracil-DNA glycosylase inhibitor (UGI). These inhibitors suppress uracil excision by UNG, thereby favoring the C•G to T•A outcome. On the other hand, the more recently developed C•G to G•C base editors (CGBEs) utilize a variety of different DNA repair effectors, including but not limited to UNG, X-Ray Repair Cross Complementing 1 (XRCC1), and RNA-binding motif protein X-linked (RBMX)[5–7]. These effectors serve to bias the cellular repair processes towards the C•G to G•C outcome. Studies have revealed that the sequence context surrounding the uracil intermediate greatly influences these two outcomes as well[7,8].

Prior work to investigate cytosine base editing mechanisms have utilized Repair-seq screens, in which the editing site is directly sequenced in the same amplicon as the sgRNA used for gene

[1]Department of Chemistry and Biochemistry, University of California, San Diego, CA, USA. [2]Moores UCSD Cancer Center, University of California, San Diego, CA, USA. [3]Sanford Stem Cell Institute, University of California, San Diego, CA, USA. [4]These authors contributed equally: Sifeng Gu, Zsolt Bodai. ✉e-mail: akomor@ucsd.edu

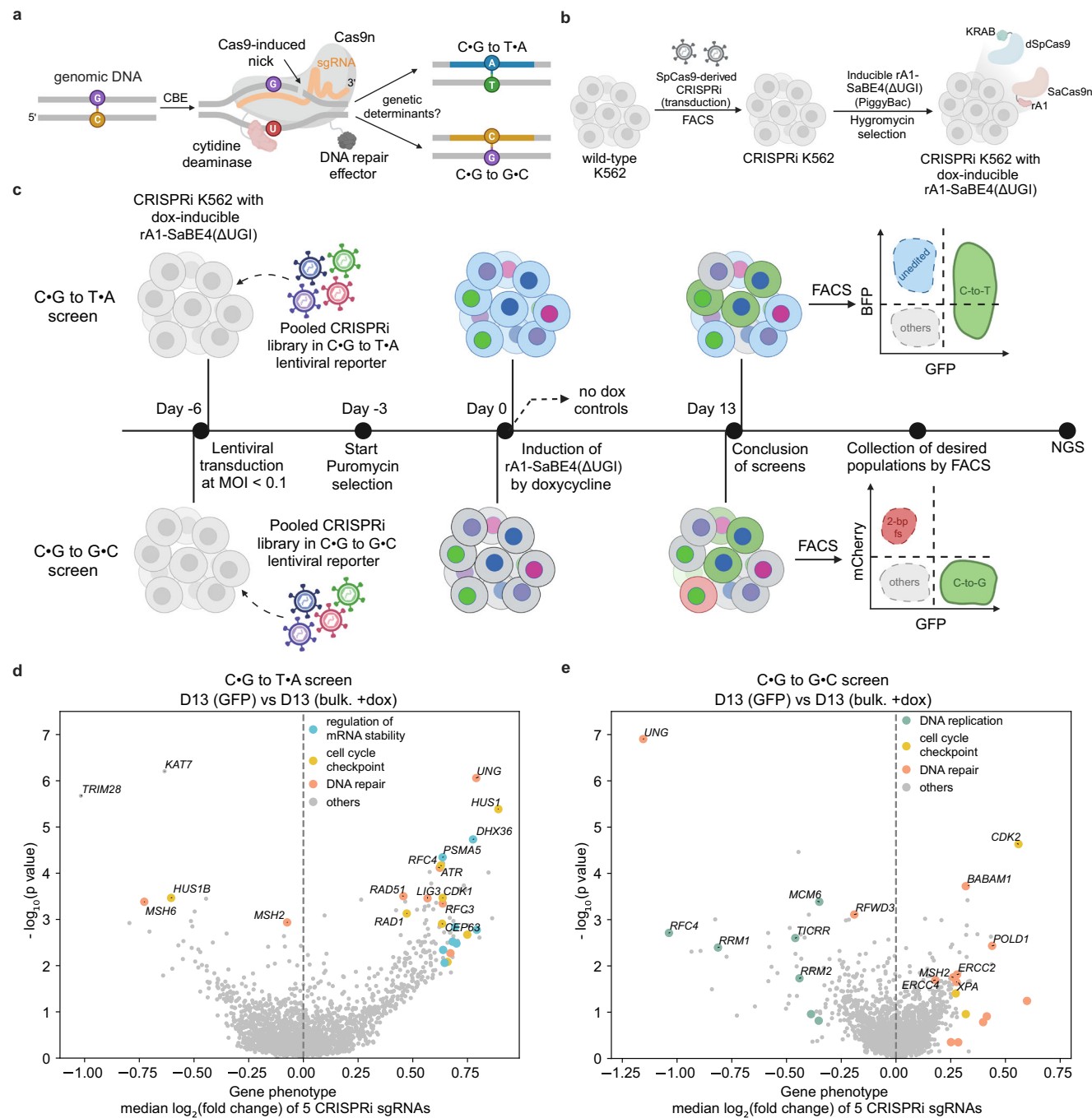

**Fig. 1 | Development and implementation of CRISPRi screens for identifying genes involved in processing cytosine base editing intermediates. a** Schematic of cytosine base editing. Binding of Cas9n to a genomic locus of interest, facilitated by base-pairing between the DNA and the sgRNA, exposes a window of single-stranded DNA to the cytidine deaminase. Cytosines within this window are deaminated to uracil, while the Cas9n cleaves the opposite strand, to produce a U•G intermediate with a nick. This intermediate is processed by the cell to produce either a C•G to T•A outcome (top), or C•G to non-T•A outcomes, the most common of which is a C•G to G•C (bottom). The genetic determinants for the distribution between the C•G to T•A and C•G to G•C outcomes are still unclear. **b** Schematic of the generation of CRISPRi-expressing K562 cell lines with integrated dox-inducible rA1-SaBE4(ΔUGI). The CRISPRi constructs shown in Supplementary Fig. 2a were transduced into K562 cells, and successfully transduced cells were selected using fluorescence-activated cell sorting (FACS). The resulting CRISPRi-expressing K562 cells were then transfected with the dox-inducible rA1-SaBE4(ΔUGI) construct shown in Supplementary Fig. 2a and a PiggyBac transposase. Cells with rA1-SaBE4(ΔUGI) successfully integrated were selected for with a hygromycin treatment. **c** Timeline of the C•G to T•A and C•G to G•C screens. For both screens, the cell line described in (**b**) was transduced with the lentiviral reporter vectors described in Supplementary Fig. 2b at a low multiplicity of infection (MOI < 0.1). Volcano plots from the C•G to T•A (**d**) and C•G to G•C (**e**) screens are shown. Representative genes from enriched pathways identified via GOBP analysis are highlighted in colors indicated in the figure legends. Each screen was conducted in $n = 2$ independent replicates. Gene phenotypes (median $\log_2$ fold change of 5 CRISPRi sgRNAs) and $P$ values were calculated using the MAGeCK RRA. MAGeCK uses two-sided negative binomial tests with FDR correction at the sgRNA level and one-sided permutation-adjusted RRA tests at the gene level. Source data are provided as a Source Data file. Created in BioRender. Gu, S. (2025) https://BioRender.com/b8hi5u8.

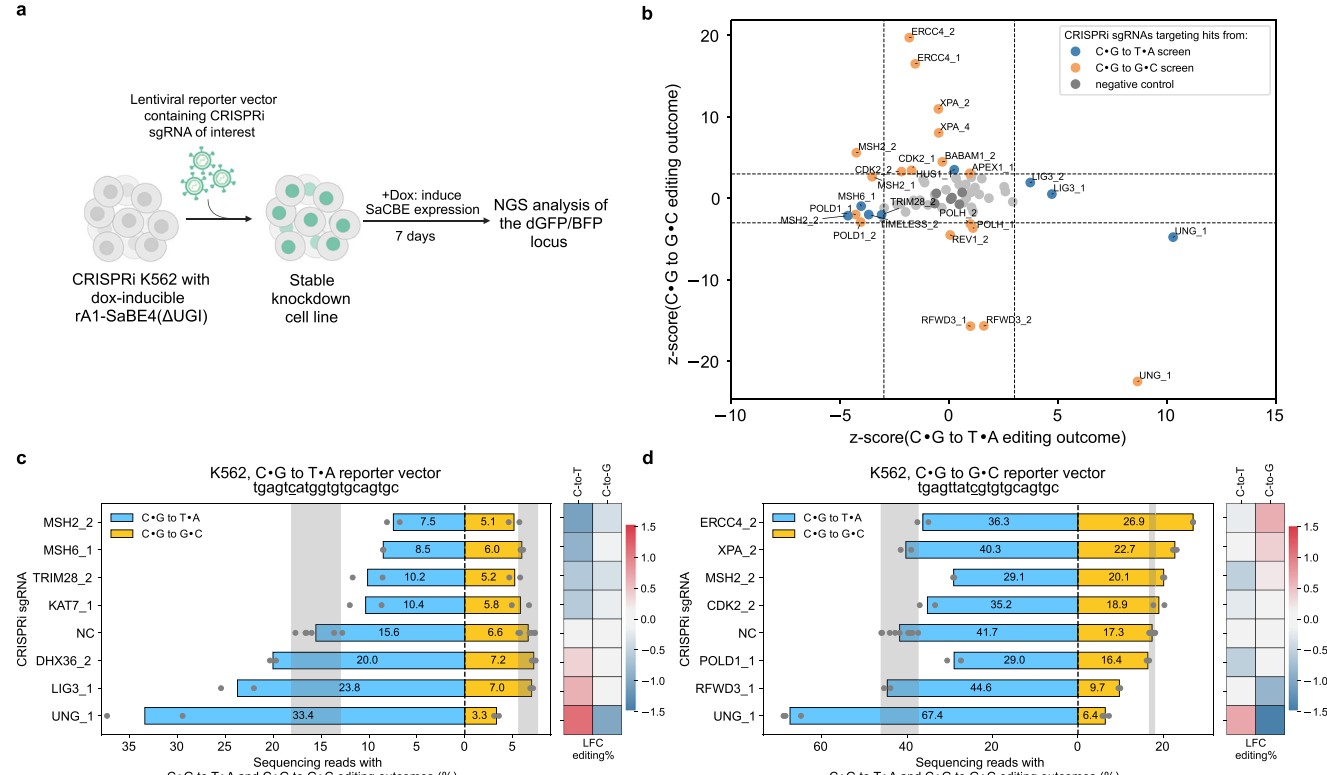

**Fig. 2 | CRISPRi screen results and validation. a** Schematic of hit validation strategy. The CRISPRi Sp-sgRNAs from Supplementary Fig. 6a, b were individually cloned into one of the constructs from Supplementary Fig. 2b (depending on which screen the potential gene hit was identified from). These were then individually transduced into the cell lines from Fig. 1b to produce stable knockdown cell lines for genes of interest. Doxycycline was added to induce rA1-SaBE4(ΔUGI) expression, and base editing was allowed to occur for 7 days. The *BFP* or *dGFP* locus (site of editing) was then sequenced with NGS to quantify editing efficiencies. Created in BioRender. Gu, S. (2025) https://BioRender.com/7923b17. **b** Z-scores of changes in C•G to T•A editing efficiencies for each CRISPRi Sp-sgRNA (x-axis) were plotted against the z-scores of changes in C•G to G•C editing efficiencies (y-axis). For both screens, sgRNAs that have |z-scores| > 3 in either the C•G to T•A or C•G to G•C editing outcome are considered to have strong effects and are highlighted in color

corresponding to their respective screen from which they were identified as a hit. Absolute C•G to T•A and C•G to G•C editing efficiencies at the single-copy, genomically-integrated C•G to T•A (**c**) and C•G to G•C (**d**) reporters upon knockdown of selected genes with the best-performing CRISPRi sgRNAs are shown. Negative control (NC) samples are also shown, in which cells were treated identically using non-targeting sgRNAs. For the NC samples, the values represent the mean obtained from 3 different non-targeting sgRNAs, with $n = 3$ biological replicates each. For all other samples, the values represent the mean of $n = 2$ biological replicates. The shaded regions represent the ranges that are within three STDs from the mean C•G to T•A and C•G to G•C editing efficiencies of the NC samples. Heatmaps represent $\log_2$ (fold change) (LFC) in C•G to T•A and C•G to G•C editing efficiencies relative to the corresponding means of non-targeting sgRNAs. Source data are provided as a Source Data file.

knockdown[7,9]. These screens were performed in HeLa or HEK293T cells, utilized transient transfection of the base editor, and the target site was a non-transcribed locus with multiple target Cs. These studies found that UNG and the E3 ubiquitin ligase RFWD3 had the largest impact on the C•G to G•C outcome upon knockdown and found UNG and DSB repair factors to be involved in the indel outcome. Overall, beyond the involvement of UNG in producing C•G to non-T•A outcomes, a detailed explanation of the genetic mechanisms governing cytosine base editing outcomes is still lacking[10]. In particular, we were interested to identify the cellular repair factors involved in producing the desired C•G to T•A outcome.

We reasoned that coupling gene knockdown with fluorescent reporters for the two major cytosine base editing outcomes could further illuminate the underlying genetic mechanisms governing these outcomes. We therefore developed and executed two CRISPR-interference (CRISPRi) screens in K562 cells that measured the impact of knockdown of 2015 individual genes on cytosine base editing activity at genomically integrated, actively transcribed, C•G to T•A and C•G to G•C fluorescent reporters in which only a single target C was edited. We additionally comprehensively analyzed and validated the hits from our screens across multiple target sites and cell lines. We find that MutSα (the MSH2/MSH6 heterodimer), the mismatch repair (MMR) recognition complex, is involved in

facilitating C•G to T•A outcomes. While previous work on APOBEC-mediated carcinogenesis has identified MMR as a source of mutagenesis[11], its role in cytosine base editing has not been demonstrated previously. We also find that RFWD3, an E3 ligase, mediates a translesion synthesis pathway that specifically leads to the C•G to G•C outcome (and not the C•G to A•T outcome)[12,13]. Finally, we show that XPF (encoded by *ERCC4*), a 3′-flap endonuclease, is involved in repairing the intermediate back to the original C•G base pair. This study fills in several key knowledge gaps in our understanding of cytosine base editing.

## Results

### Development of fluorescent reporters for CBE outcomes

To systematically identify enzymes that affect cytosine base editing outcomes, we coupled fluorescent reporters for C•G to T•A or C•G to G•C editing with CRISPRi screens[14]. Our C•G to T•A fluorescent reporter leveraged a blue fluorescent protein (BFP) variant that is converted into green fluorescent protein (GFP) upon a single C•G to T•A conversion (Supplementary Fig. 1a)[15,16] Our C•G to G•C fluorescent reporter utilized a non-fluorescent GFP variant that is converted into functional GFP upon a single C•G to G•C conversion (Supplementary Fig. 1b). We also incorporated a frameshifted mCherry into this reporter construct to report on indel formation[17]. Notably, we codon optimized both

constructs within the protospacer sequences to remove any bystander cytosines, ensuring only one target C lies within the editing window.

To test our reporter systems, HEK293T cells were transfected with plasmids encoding one of the reporters, a SaBE4(ΔUGI) construct (a cytidine deaminase fused to the PAM-relaxed KKH *Staphylococcus aureus* Cas9n[18] via an optimized 32 amino acid linker[19]), and a sgRNA targeting the CBE to the mutation of interest (Supplementary Fig. 1a, b). In this initial round of experiments, we evaluated three different SaBE4(ΔUGI) constructs, in which the deaminase domain was rA1 (APOBEC1 from *Rattus norvegicus*), eA3A (an engineered human APOBEC3A) or RrA3F (APOBEC3F from *Rhinopithecus roxellana*)[3,20,21]. Further, we used SaCas9-derived CBEs to avoid sgRNA exchange between the CBE and SpCas9-derived CRISPRi machinery in future experiments. After 72 h, cells were harvested and analyzed by flow cytometry for either BFP to GFP conversion (for the C•G to T•A reporter) or GFP turn-on (for the C•G to G•C reporter)[20,21]. We found that rA1 outperformed both the RrA3F and eA3A on both reporters. The rA1-derived CBE facilitated GFP fluorescence turn-on in 45.2% and 13.6% of cells for the C•G to T•A and C•G to G•C reporters, respectively, compared to 40.4% and 7.7% for the RrA3F-derived CBE, and 25.2% and 8.4% for the eA3A-derived CBE (Supplementary Fig. 1c, d). We observed no mCherry fluorescence turn-on with the C•G to G•C reporter, which we attribute to plasmid degradation rather than repair following in situ DSB introduction. We therefore used the rA1-derived SaBE4(ΔUGI) construct, hereafter referred to as rA1-SaBE4(ΔUGI), for subsequent experiments.

## Coupling fluorescent reporters for CBE outcomes with CRISPRi screens

To develop the two screening systems, we first generated two stable CRISPRi-expressing K562 cell lines (one for each screen, Fig. 1b and Supplementary Fig. 2a) via lentiviral transduction[14,22]. We then incorporated doxycycline-inducible rA1-SaBE4(ΔUGI) into both cell lines via PiggyBac transposition (Supplementary Fig. 2a). Finally, we generated lentiviral reporter vectors containing the C•G to T•A or C•G to G•C reporter, a Sp-sgRNA cassette encoding the CRISPRi Sp-sgRNA library, and a Sa-sgRNA cassette guiding the CBE to the reporter (Supplementary Fig. 2b). The CRISPRi Sp-sgRNA library is detailed in Supplementary Fig. 2c and Supplementary Data 1.

We conducted the screens, each in duplicate, according to Fig. 1c (see Methods). In the doxycycline-treated cells, we observed on average 13.2 ± 0.4% of cells with GFP fluorescence (C•G to T•A outcome) in the C•G to T•A screens, and 12.9 ± 1.7% of cells with GFP fluorescence (C•G to G•C outcome) and 2.0 ± 0.1% of cells with mCherry fluorescence (C•G to +2 frameshift indel outcome) in the C•G to G•C screens (Supplementary Fig. 3a, b). We then used fluorescence-activated cell sorting (FACS) to sort a subset of the doxycycline-treated cells into the various populations of interest. Unfortunately, the number of cells collected from the mCherry+ population in the C•G to G•C screens was too low to obtain enough sgRNA coverage to acquire reliable data on sgRNA enrichment or depletion in this population. Unsorted, doxycycline-treated cells, and untreated cells served as bulk population controls.

We validated the screens by first sequencing the *BFP* (C•G to T•A screen) or *dGFP* (C•G to G•C screen) loci of the unsorted and sorted populations and observed both stringent doxycycline induction and significant enrichment of the editing outcomes of interest in the appropriate sorted populations (Supplementary Fig. 3c, d and Supplementary Discussion). We then amplified and sequenced the CRISPRi Sp-sgRNA regions of each population of cells with larger than 200X sequencing depth using next generation sequencing (NGS) (Supplementary Fig. 4a, d). As an initial validation of the screen, we monitored dropout of essential genes and found effective gene knockdown in both screens (Supplementary Fig. 4e, h and Supplementary Discussion). We additionally analyzed the data to identify genes that are

synthetic lethal to CBE expression and/or activity (i.e., genes whose knockdown causes preferential cell death in cells in which CBE is expressed) and notably did not observe any pathways that were significantly enriched upon CBE expression (See Supplementary Discussion).

## Identification of genes governing the C•G to T•A and C•G to G•C editing outcomes

To identify genes influencing C•G to T•A and C•G to G•C outcomes, we compared the distributions of the CRISPRi Sp-sgRNAs in the GFP+ populations to that of their respective pre-sorted bulk populations (bulk, +dox samples) for both screens. Depending on the screen, genes whose sgRNAs are depleted in the GFP+ population are expected to be those that promote the C•G to T•A or C•G to G•C outcome, whereas genes whose sgRNAs are enriched in this population are expected to be those that inhibit these outcomes. We observed good correlation (Pearson's r > 0.7) between the $\log_2$ fold changes of the CRISPRi Sp-sgRNAs in the GFP+ population compared to the initial day 0 population for the two replicates from both screens (Supplementary Fig. 5a–c). However, we observed a low correlation (Pearson's r < 0.7) between the $\log_2$ fold changes of the CRISPRi Sp-sgRNAs in the GFP+ population compared to the unsorted population on day 13 (bulk +dox samples) for the two replicates (Supplementary Fig. 5b–d). We believe this is not indicative of low screen quality and is due to only a small number of genes from our library being able to single-handedly impact base editing outcomes.

We used the MAGeCK RRA (Model-based Analysis of Genome-wide CRISPR-Cas9 Knockout Robust Rank Aggregation) algorithm to identify genes whose sgRNAs were consistently depleted or enriched in the GFP+ populations (Fig. 1d, e). *UNG*, which served as our positive control, was the strongest hit in both screens. UNG is known to excise the uracil intermediate during base excision repair (BER) and initiate either conversion back to the initial C•G base pair, or conversion to the C•G to G•C outcome. Consistent with this mechanism, *UNG* was enriched in the GFP+ population in the C•G to T•A screen and depleted in the GFP+ population in the C•G to G•C screen. A gene ontology biological process (GOBP) analysis of the C•G to T•A screen results revealed that sgRNAs targeting genes involved in DNA repair, cell cycle checkpoints, and regulation of mRNA stability were enriched, and sgRNAs targeting genes encoding for the MutSα complex (*MSH2* and *MSH6*) and two chromatin remodeling enzymes (*TRIM28* and *KAT7*) were significantly depleted (Fig. 1d, full list in Supplementary Data 2). A GOBP analysis of the C•G to G•C screen results also identified sgRNAs targeting DNA repair genes, especially those involved in nucleotide excision repair (NER), were enriched, and sgRNAs targeting other DNA repair genes (*UNG* and *RFWD3*), as well as DNA replication genes were depleted (Fig. 1e, full list in Supplementary Data 3).

Taking into account *p*-values, fold-change values, and the GOPB analysis, we selected 14 top-ranked hits from the C•G to T•A screen and 9 top-ranked hits from the C•G to G•C screen for further validation (Supplementary Fig. 6a, b). We additionally included 7 genes that have been hypothesized to resolve the CBE intermediate but did not show up as hits from the screens[10]. Specifically, we included three BER genes, two error-prone Y-family polymerases, one gene involved in DNA damage check point and one NER gene. Lastly, we included 3 non-targeting sgRNA controls.

## Initial validations suggest C•G to T•A editing relies on MSH2 and MSH6 and is hindered by LIG3 and UNG

For each gene chosen for validation, we chose the two best-performing CRISPRi Sp-sgRNAs, as determined by the two largest $\log_2$ fold changes in the screening results. We individually incorporated each CRISPRi Sp-sgRNA into either the C•G to T•A or C•G to G•C lentiviral reporter vector, depending on from which screen the gene was a hit. To generate stable knockdown cell lines for each gene of interest, we then

individually transduced these vectors into the CRISPRi K562 cell line with dox-inducible rA1-SaBE4(ΔUGI) at a MOI < 0.3. We enriched for transduced cells with a puromycin selection, and then added doxycycline to induce rA1-SaBE4(ΔUGI) expression for 7 days. Editing efficiencies at the *BFP/dGFP* loci were quantified with NGS (Fig. 2a and Supplementary Figs. 7, 8). We then computed a z-score for each CRISPRi Sp-sgRNA relative to the non-targeting sgRNA controls, with sgRNAs that demonstrated absolute z-scores > 3 considered to have strong impacts on editing outcomes (Fig. 2b).

Out of the hits from the C•G to T•A screen, we found that knockdown of *MSH2* and *MSH6* led to the most substantial decreases in C•G to T•A editing efficiencies, whereas knockdown of *LIG3* and *UNG* caused the most significant increases in C•G to T•A editing efficiencies (Fig. 2b, c and Supplementary Fig. 7). The C•G to T•A and C•G to G•C editing efficiencies of the C•G to T•A reporter vector in the non-targeting CRISPRi Sp-sgRNA samples averaged 15.6 ± 1.7% and 6.6 ± 0.7%, respectively. Knockdown of *MSH2* and *MSH6* decreased C•G to T•A editing efficiencies by 2.1 ± 0.3-fold (to 7.5 ± 0.7%) and 1.8 ± 0.2-fold (to 8.5 ± 0.1%), respectively, while maintaining C•G to G•C editing efficiencies at 5.1 ± 0.6% and 6.0 ± 0.1%, respectively. Meanwhile, knockdown of *LIG3* increased C•G to T•A editing efficiency by 1.5 ± 0.2-fold (to 23.8 ± 1.7%) while maintaining C•G to G•C editing efficiency at 7.0 ± 0.1%. *UNG* knockdown robustly increased C•G to T•A editing efficiency by 2.1 ± 0.3-fold and decreased C•G to G•C editing efficiency by 2.0 ± 0.3-fold, as expected. These findings are consistent with a mechanism in which the C•G to T•A outcome is facilitated by MSH2 and MSH6, and inhibited by LIG3 and UNG.

In addition, we found that knockdown of *TRIM28*, a transcriptional repressor, and *KAT7*, a histone acetyltransferase, both decreased C•G to T•A editing efficiencies by around 1.5-fold while knockdown of *DHX36*, a DNA/RNA guanine-quadruplex helicase, increased the C•G to T•A editing efficiency by 1.3 ± 0.1-fold. These genes participate in multiple DNA repair pathways as well as additional biological processes such as chromatin remodeling and regulation of mRNA stability[23–25]. Their exact roles in processing base editing intermediates are likely multifaceted and may involve influencing base editor expression levels and/or impacting accessibility of the target site.

### Initial validations suggest C•G to G•C editing depends on RFWD3 and UNG and is inhibited by XPF and XPA

Out of the hits from the C•G to G•C screen, we found that knockdown of *ERCC4* and *XPA* increased C•G to G•C editing efficiencies most significantly, while knockdown of *RFWD3* and *UNG* decreased C•G to G•C editing efficiencies most significantly (Fig. 2b, d, and Supplementary Fig. 8). The C•G to T•A and C•G to G•C editing efficiencies of the C•G to G•C reporter vector in the non-targeting CRISPRi Sp-sgRNA samples averaged 41.7 ± 3.0% and 17.3 ± 0.5%, respectively. Knockdown of *ERCC4* and *XPA* increased C•G to G•C editing efficiencies by 1.6 ± 0.1-fold (to 26.9 ± 0.1%) and 1.3 ± 0.1-fold (to 22.7 ± 0.4%), respectively. XPF and XPA (the proteins encoded by *ERCC4* and *XPA*, respectively), together with ERCC1 are known to form a ternary protein complex that acts as a structure-specific endonuclease, specifically cleaving the DNA backbone at junctions between double-stranded and single-stranded DNA[26]. However, we did not observe knockdown of *ERCC1* to significantly affect either the C•G to G•C or C•G to T•A outcome (Supplementary Fig. 8). Knockdown of *RFWD3* decreased C•G to G•C editing efficiency by 1.8 ± 0.1-fold (to 9.7 ± 0.2%), while maintaining C•G to T•A editing efficiency at 44.7 ± 1.1%. Knockdown of *UNG* decreased C•G to G•C editing efficiency by 2.7 ± 0.2-fold (to 6.4 ± 0.5%) and increased C•G to T•A editing efficiency by 1.6 ± 0.1-fold (to 67.4 ± 1.3%). These results are consistent with the C•G to G•C outcome being dependent on *RFWD3* and *UNG* and inhibited by *ERCC4* and *XPA*.

Unexpectedly, while both *MSH2* and *POLD1* CRISPRi Sp-sgRNAs were enriched in the GFP+ population in the C•G to G•C screen, their knockdown primarily impacted C•G to T•A editing, leading to a 1.4-fold

decrease in C•G to T•A editing efficiency. By decreasing the absolute C•G to T•A editing efficiencies while maintaining the C•G to G•C editing efficiencies, these two gene knockdowns led to increases in the relative C•G to G•C outcome (Supplementary Fig. 9a, b). Specifically, the average relative percent of edited reads with a C•G to G•C outcome was 23.7 ± 0.2% in the negative control sgRNA samples, which increased to 30.9 ± 0.1% upon *MSH2* knockdown, and 27.6 ± 0.9% upon *POLD1* knockdown. The effect of *MSH2* knockdown on editing outcomes was consistent with both reporters. Importantly, these data reveal that the C•G to T•A outcome is also dependent on *POLD1*, which encodes for the catalytic subunit of the replicative polymerase δ that is involved in MMR[27].

Because we quantified editing with NGS, we were able to also evaluate the impact of each selected gene knockdown on the C•G to A•T outcome, which is the least frequent cytosine base editing outcome (Supplementary Fig. 10). UNG knockdown reduced both the C•G to G•C and C•G to A•T editing efficiencies by 2.7 ± 0.2-fold and 3.4 ± 0.3-fold, respectively (Supplementary Fig. 10b), implying that both outcomes rely on abasic site formation via UNG-mediated uracil excision. Surprisingly, while knockdown of *RFWD3* caused a 1.8 ± 0.1-fold decrease in C•G to G•C editing efficiency, its knockdown caused a modest 1.2 ± 0.1-fold increase in C•G to A•T editing efficiency (Supplementary Fig. 10b)[12,13]. The contrasting effects of *RFWD3* knockdown on these two outcomes suggest the existence of distinct translesion synthesis sub-pathways, with RFWD3 promoting the one that specifically leads to the C•G to G•C outcome. Knockdown of *ERCC4* and *XPA* elevated both the C•G to G•C and C•G to A•T editing efficiencies simultaneously (Supplementary Fig. 10b), consistent with an inhibitory role of XPF and XPA in these outcomes.

### Selected genes were further validated at two endogenous sites in K562 cells

To further validate these hits beyond the transduced lentiviral reporters (the *BFP* and *dGFP* genes), we chose seven genes, *UNG*, *MSH2*, *MSH6*, *POLD1*, *LIG3*, *RFWD3* and *ERCC4*, to validate at two endogenous sites, *HEK3* and *RNF2*. We replaced the *BFP*-targeting Sa-sgRNA spacer in the C•G to T•A lentiviral reporter vector with either the *HEK3*- or *RNF2*-targeting spacer sequences. We then individually incorporated the best CRISPRi Sp-sgRNA for each gene and three non-targeting sgRNAs into the vector and separately transduced each sgRNA-expressing vector into the CRISPRi K562 cell line with dox-inducible rA1-SaBE4(ΔUGI) at a MOI < 0.3. Editing at these two endogenous sites was allowed for 7 days before the *HEK3* or *RNF2* loci were amplified and quantified by NGS. Additionally, knockdown levels of the target genes in these cell lines were quantified by RT-qPCR (Supplementary Fig. 11a).

Overall, we observed consistent trends with our previous observations at the *GFP* reporters. Knockdown of MMR genes (*MSH2*, *MSH6*) and *POLD1* reduced C•G to T•A editing efficiencies at both sites. *POLD1* knockdown led to the most significant (1.4 ± 0.2-fold) decrease in absolute C•G to T•A editing at the *HEK3* site, while *MSH2* knockdown resulted in the most substantial (1.6 ± 0.1-fold) reduction in absolute C•G to T•A editing efficiency at the *RNF2* site (Fig. 3a, b). In contrast, knockdown of *LIG3* increased the absolute C•G to T•A editing efficiencies by around 1.4-fold at both the *HEK3* and *RNF2* sites. Relative C•G to T•A outcomes (normalized to total editing) at both sites were impacted similarly upon knockdown of these four genes (Supplementary Fig. 11b, c).

Knockdown of *RFWD3* did not lead to a substantial decrease in absolute C•G to G•C editing efficiency at the *HEK3* site, but its knockdown did increase both C•G to T•A and C•G to A•T editing efficiencies by 1.3 ± 0.2-fold and 1.5 ± 0.2-fold, respectively. At the *RNF2* site, *RFWD3* knockdown significantly decreased the absolute C•G to G•C editing efficiency by 1.9 ± 0.1-fold, while also promoting both C•G to T•A and C•G to A•T editing efficiencies by around 1.2-fold (Fig. 3a, b). As a result, *RFWD3* knockdown selectively suppressed the relative C•G to

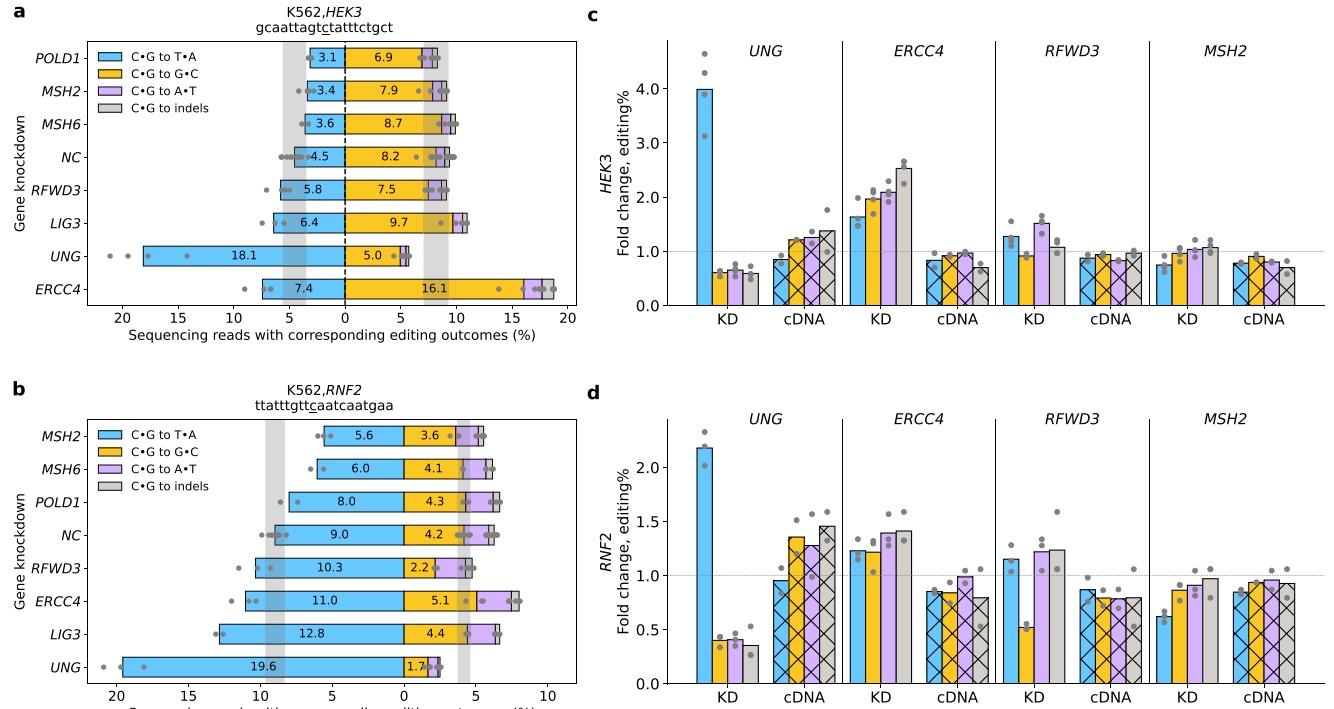

**Fig. 3 | Validation of selected genes at two endogenous sites in K562. a, b** The dCas9-BFP-KRAB and dox-inducible rA1-SaBE4(ΔUGI) expressing K562 cell line (Fig. 1b) was transduced with the C•G to T•A lentiviral reporter vector (Supplementary Fig. 2b), in which the *BFP*-targeting Sa-sgRNA spacer was replaced with either a *HEK3*- or *RNF2*-targeting spacer, and the CRISPRi Sp-sgRNA was targeted to the specific gene indicated. After a 3-day puromycin selection, doxycycline was added and rA1-SaBE4(ΔUGI) expression was induced for 7 days. The *HEK3* or *RNF2* locus (site of editing) was then sequenced with NGS to quantify editing efficiencies. Editing efficiencies of all outcomes at the two endogenous sites, *HEK3* (**a**) and *RNF2* (**b**), upon knockdown of selected genes are shown. Negative control (NC) samples are also shown, in which cells were treated identically using non-targeting sgRNAs. For the NC samples, the bars represent the mean obtained from 3 non-targeting sgRNAs, with *n* = 4 biological replicates each. For all other samples, the bars represent the mean of *n* = 2 (*MSH6* and *POLD1*), 3 (*LIG3*), or 4 (*UNG*, *MSH2*, *RFWD3*,

and *ERCC4*) biological replicates. The shaded regions represent the ranges that are within three STDs from the mean C•G to T•A and C•G to G•C editing efficiencies of the NC samples. **c, d** Cells were treated as described in (**a, b**), but with the *BFP* gene of the reporter construct replaced with cDNA encoding for the corresponding knocked-down gene. Shown are the fold-changes in editing efficiencies of all outcomes upon knockdown of selected genes at the two endogenous sites, *HEK3* (**c**) and *RNF2* (**d**), in comparison to the average editing efficiencies of 3 negative control samples without knockdown. The gene knockdown effects on all editing outcomes (solid color bars) at both sites were fully or partially rescued by the re-expression of the corresponding protein (striped bars). The bars represent the mean fold-changes for *n* = 4 and *n* = 3 biological replicates for the KD condition at *HEK3* and *RNF2* site, respectively, and *n* = 2 biological replicates for cDNA re-expression condition. KD, knockdown. Source data are provided as a Source Data file.

G•C outcome while promoting all relative C•G to non-G•C outcomes at both sites (Supplementary Fig. 11b, c). Knockdown of *ERCC4* increased all outcomes at both the *HEK3* and *RNF2* sites (Fig. 3a, b). This effect was most notable at the *HEK3* site, in which C•G to T•A, C•G to G•C, C•G to A•T, and C•G to indels all increased by 1.6 ± 0.3-fold, 2.0 ± 0.2-fold, 2.1 ± 0.2-fold, and 2.5 ± 0.3-fold, respectively. As a result, the total editing efficiency increased by 1.9 ± 0.2-fold, suggestion an active role of XPF in repairing the base editing intermediate at *HEK3*.

Knockdown of *UNG* increased absolute C•G to T•A editing efficiencies by 4.0 ± 0.7-fold and 2.2 ± 0.2-fold at the *HEK3* and *RNF2* sites, respectively. Meanwhile, its knockdown significantly decreased all C•G to non-T•A outcomes (Fig. 3a, b). This agrees with the established role of *UNG* in excising the uracil intermediate. As expected, *UNG* knockdown also led to a major increase in the relative C•G to T•A outcome and decreases in all relative C•G to non-T•A outcomes (Supplementary Fig. 11b, c).

To confirm that the observed knockdown effects were due to on-target CRISPRi knockdown, we selected one gene per pathway (*UNG*, *ERCC4*, *RFWD3*, and *MSH2*, which generally caused the greatest changes in editing profiles upon knockdown) to perform re-expression rescue experiments. For each of these four genes, we replaced the *BFP* gene of the reporter constructs with the cDNA of the corresponding knocked down gene to achieve stable re-expression in the appropriate CRISPRi background. Re-expression of *UNG*, *ERCC4*, *RFWD3*, and *MSH2*

all fully or partially rescued the knockdown effects on all editing outcomes at both the *HEK3* and *RNF2* sites (Fig. 3c, d).

## MutSα and UNG compete to process cytosine base editing intermediates

To further explore how interplays between these genes influence different editing outcomes, we transfected the CRISPRi K562 stable knockdown cell lines (from Fig. 2a; in which *UNG*, *MSH2*, or *MSH6* were knocked down, see Methods) with either rA1-SaBE4(ΔUGI), rA1-SaBE4 (which contains two copies of UGI and thus inhibits UNG during the process of base editing), or rA1-SaBE1 (which uses dSaCas9 and thus does not introduce a nick on the non-edited strand), and measured their editing outcomes at the *HEK3* and *RNF2* sites (Supplementary Fig. 12a). This approach allowed us to dissect the contributions of UNG and the Cas9-induced nick to the various editing outcomes. Compared to the integrated rA1-SaBE4(ΔUGI) system, transfection with rA1-SaBE4(ΔUGI) demonstrated faster editing rates, with significantly higher overall editing efficiencies in 3 days (Supplementary Fig. 12b, c). Moreover, we observed bystander editing at the C16 and C14 positions of the *HEK3* and *RNF2* protospacers, respectively, in addition to the C10 position. Notably, The C10 position was the only edited position when using the integrated rA1-SaBE4(ΔUGI) system, and it is highly likely that multi-uracil intermediates are processed through different mechanisms. Because UGI inhibits endogenous UNG, editing with rA1-

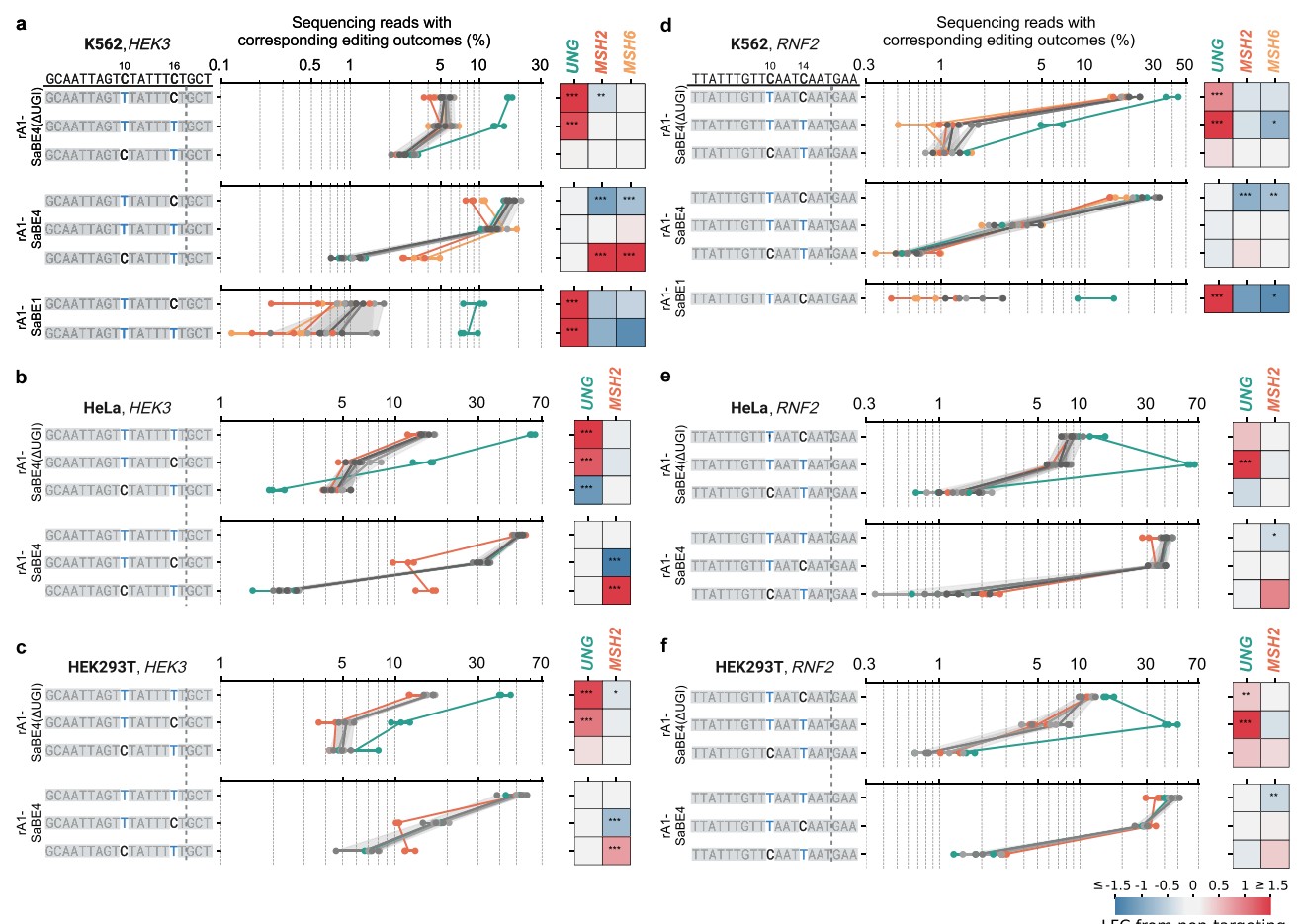

**Fig. 4 | Impacts of *MSH2*, *MSH6* and *UNG* knockdown on C•G to T•A outcomes in K562, HeLa, and HEK293T cells.** Cells were treated as described in Supplementary Fig. 12a (K562), 13a-b (HeLa), or 14a-c (HEK293T) to induce knockdown of *UNG*, *MSH2*, or *MSH6*. Cells were then transfected with plasmids encoding rA1-SaBE4(-ΔUGI), rA1-SaBE4, or rA1-SaBE1 [rA1-dSaCas9(ΔUGI)] and Sa-gRNA targeting the *HEK3* or *RNF2* loci. Three days post-transfection, transfected cells were collected by FACS and editing efficiencies at the target sites were quantified by NGS. Effects of indicated gene knockdown on C•G to T•A editing outcomes at the *HEK3* site in K562 (**a**), HeLa (**b**) and HEK293T (**c**) cells are shown. Effects of indicated gene knockdown on C•G to T•A editing outcomes at the *RNF2* site in K562 (**d**), HeLa (**e**) and HEK293T (**f**) cells are shown. Protospacer sequences are shown at the top. All C•G to T•A editing outcomes at these two sites are listed, and edits are highlighted in color.

Vertical dashed lines mark the Cas9-induced nick position. Base editors used are labeled on the left. Tract plots display absolute frequencies of editing outcomes upon specific gene knockdown (color-coded by gene as indicated). Three negative control samples without knockdown are shown in gray, with shaded areas representing ±2 STDs of mean. Dots show individual replicates, and tract lines show averages of $n = 2$ (**d**: *NCs*, *UNG* and *MSH2*) or $n = 3$ (all the rest) biological replicates. Horizontal lines show ranges of the replicates for corresponding outcomes. Heatmaps show log$_2$ (fold change) (LFC) in frequencies of editing outcomes relative to the corresponding means of negative controls. *: 1.5 <adjusted z-score <2. **: 2<adjusted z-score <3. ***: adjusted z-score > 3. Source data are provided as a Source Data file.

SaBE4 predominantly yielded C•G to T•A outcomes. Editing efficiencies with rA1-SaBE1 were much lower than that of the rA1-SaBE4(ΔUGI) system in the negative control cells (5.6 ± 1.5-fold and 13.0 ± 2.9-fold at the C10 position of *HEK3* and *RNF2* sites, respectively), highlighting the importance of the Cas9-induced nick for base editing.

Knockdown of *MSH2* and *MSH6* in this new system with the rA1-SaBE4(ΔUGI) editor did not lead to a substantial reduction in the single C•G to T•A editing outcome at the C10 positions (Fig. 4a, d). However, we will note that due to the elevated rate of editing at the C14 and C16 positions in this system, the single C•G to T•A outcomes could be derived from dual uracil intermediates, which adds complexity to this observation. Interestingly, the effects of *MSH2* or *MSH6* knockdown were pronounced when UNG was simultaneously inhibited via the use of the rA1-SaBE4 construct. Specifically, when editing with rA1-SaBE4, knockdown of either *MSH2* or *MSH6* led to significant decreases in the frequencies of the single C•G to T•A editing outcome at the C10 position at both the *HEK3* and *RNF2* sites (Fig. 4a, d). At the *HEK3* site, knockdown of *MSH2* or *MSH6* reduced the absolute editing efficiency

of this outcome by 2.1 ± 0.2-fold or 1.6 ± 0.1-fold, respectively. At the *RNF2* site, knockdown of *MSH2* or *MSH6* reduced the absolute editing efficiency of this outcome by 1.9 ± 0.2-fold or 1.6 ± 0.3-fold, respectively. Because MutSα and UNG both recognize the U•G mismatch, a potential explanation for these observations is that MutSα and UNG may compete for substrate recognition, with UNG having a competitive advantage. Two potential factors may synergize to give UNG such a competitive advantage over MutSα in the transfection system. First, multi-uracil intermediates are not optimal substrates for MutSα[28]. In addition, multiple uracils could enhance the recruitment of UNG to the editing site.

Intriguingly, we found the effect of MutSα knockdown to be dependent on the target base position when editing with rA1-SaBE4. At the *HEK3* site, knockdown of *MSH2* or *MSH6* had minimal impact on the dual C•G to T•A outcome frequency, but led to a 2.9 ± 0.8-fold or 3.8 ± 1.0-fold *increase* in the frequency of the single C•G to T•A editing outcome at the C16 position, respectively (Fig. 4a). Since the C16 position is only one base pair away from the Cas9-induced nick, it may

be processed productively (to produce a C•G to T•A outcome) via an additional pathway independent of MutSα recognition. When the target C is shifted two base pairs away from the nick to the C14 position, as in *RNF2*, knockdown of *MSH2* or *MSH6* did not result in a significant increase in the single C•G to T•A editing frequency at this position (Fig. 4d).

Detecting decreases in editing efficiencies was challenging with the rA1-SaBE1 editor due to its low editing efficiencies, but we found that *MSH6* knockdown caused a reduction in the single C•G to T•A editing outcome at the C10 position at the *RNF2* site, from 1.9 ± 0.8% to 0.8 ± 0.1% (representing a 2.5 ± 0.8-fold decrease, Fig. 4d).

The effects of *MSH2* knockdown were further validated in both HeLa and HEK293T cells with the rA1-SaBE4(ΔUGI) and rA1-SaBE4 constructs (Fig. 4b, c, e, f, Supplementary Figs. 13, 14; see Methods). In both cell types, we again observed that knockdown of *MSH2* did not (or had a minimal) impact C•G to T•A outcomes when using rA1-SaBE4 (ΔUGI). However, when editing with rA1-SaBE4, *MSH2* knockdown reduced the frequency of the single C•G to T•A editing outcome at the C10 position of *HEK3* by 2.8 ± 0.4-fold and 1.7 ± 0.1-fold in HeLa and HEK293T cells, respectively. Meanwhile, we again observed an increase in the frequency of the single C•G to T•A editing outcome at the C16 position of *HEK3* by 6.6 ± 0.4-fold and 1.7 ± 0.3-fold in HeLa and HEK293T cells, respectively (Fig. 4b, c). At the *RNF2* site, the dominant outcome was the dual C•G to T•A outcome, which was observed with 42.0 ± 0.2% and 46.8 ± 2.7% efficiencies in the negative control samples in HeLa and HEK293T cells, respectively (Fig. 4e, f and Supplementary Figs. 13e, 14e). This is compared to 3.3 ± 0.7% in K562 cells (Fig. 4d and Supplementary Fig. 12c). *MSH2* knockdown had a minimal impact on editing at this site, with the largest change being a modest reduction in the dual C•G to T•A editing frequency, of 1.3 ± 0.2-fold and 1.4 ± 0.2-fold in HeLa and HEK293T cells, respectively. We did not observe a statistically significant impact on the editing rate of the single C•G to T•A editing outcome at the C10 position in these cell lines. We believe this discrepancy may be due to the >10-fold higher editing rate of the C14 position in the HeLa and HEK293T cells compared to K562 cells.

It is noteworthy that although HEK293T cells are deficient in MMR due to the silencing of the MutLα complex (a heterodimer of MLH1 and PMS2, which is responsible for generating DNA nicks following mismatch recognition), they have normal expression levels of MutSα[29]. Interestingly, editing efficiencies between HEK293T and HeLa cells (Supplementary Fig. 13d, e and Supplementary Fig. 14d, e) are comparable, and similar impacts of *MSH2* knockdown were observed across these cell lines. A possible explanation is that mismatch recognition is required, but downstream nicking by MutLα is dispensable for cytosine base editing when using Cas9n-derived CBEs. These findings regarding MutSα address gaps in our current understanding of cytosine base editing mechanisms. Previous studies in HeLa and HEK293T cells using CBEs that lack UGI did not identify MutSα as impacting C•G to T•A editing, which is consistent with these observations[7,9]. This highlights the complexity of base editing mechanisms, which vary depending on different editors, target positions, and editing kinetics.

## RFWD3 promotes the C•G to G•C outcome via RPA-binding and ubiquitination

We next repeated the same transient transfection experiments in K562, HeLa, and HEK293T cells with *RFWD3* knockdown and the rA1-SaBE4(ΔUGI) editor. We observed consistent impacts on C•G to G•C editing outcomes by rA1-SaBE4(ΔUGI) upon *RFWD3* knockdown across K562, HeLa, and HEK29T cells (Fig. 5a–f). At the *RNF2* site, *RFWD3* knockdown reduced the frequencies of all editing outcomes containing a C•G to G•C conversion, mirroring the impact of *UNG* knockdown (Fig. 5d–f). Interestingly, this included both single C•G to G•C editing outcomes (at the C10 and C14 positions), as well as mixed C•G to G•C and C•G to T•A or A•T outcomes. At the *HEK3* site, we observed the

impact of *RFWD3* knockdown to be dependent on the position of the target base. For example, *RFWD3* knockdown did not significantly affect the frequency of the single C•G to G•C editing outcome at the C10 position (Fig. 5a–c) in any of the three cell lines. However, it reduced the frequency of the single C•G to G•C editing outcome at the C16 position by 2.2 ± 0.7-fold and 2.0 ± 0.4-fold in K562 and HEK293T cells, respectively. To further validate these findings, we repeated these experiments in HEK293T cells with four additional sgRNAs (targeting an additional site within the *RNF2* gene, one site in the *EMX1* gene, and two sites within the *HEK2* locus) using the evo-BE4(ΔUGI) (evoAPOBEC1 with SpCas9n) editor (Fig. 5g)[30]. We observed editing at a single C within the RNF2, HEK2.1, and HEK2.2 protospacers, and two Cs within the EMX1 protospacer. Across all four sites we observed robust decreases in C•G to G•C editing upon RFWD3, ranging from 1.6-fold to 1.8-fold.

Previous studies have demonstrated that RFWD3's ability to facilitate translesion synthesis depends on its RPA-binding and ubiquitination activities[12,13,31]. To determine whether RFWD3's involvement in the C•G to G•C editing outcome also relies on these functions, we co-transfected the RFWD3 knockdown HEK293T cells with plasmids encoding evo-BE4(ΔUGI), sgRNA targeting either the *RNF2* or *HEK2.1* protospacers, and one of the following RFWD3 variants: wild-type RFWD3, RFWD3[C315A], which lacks ubiquitination function, or RFWD3[I639K], which lacks RPA-binding ability (Fig. 5h). Compensation with wild-type RFWD3 substantially increased relative C•G to G•C editing efficiencies at both the *RNF2* and *HEK2.1* sites, but not the absolute C•G to G•C editing levels, which remained within error of the *RFWD3* knockdown samples. Compensation with RFWD3[C315A] decreased the absolute C•G to G•C editing efficiencies by 2.5 ± 0.8-fold and 1.3 ± 0.1-fold at the *RNF2* and *HEK2.1* sites, respectively. Meanwhile, compensation with RFWD3[I639K] did not significantly impact either the absolute or relative C•G to G•C editing efficiencies at either site. Notably, compared to RFWD3[I639K], compensation with wild-type RFWD3 decreased absolute C•G to T•A and C•G to A•T editing efficiencies at both sites. A mechanism consistent with these data is that localization of RFWD3 to the editing site may facilitate repair of the U•G intermediate to either G•C or back to C•G. Disruption of RFWD3's ubiquitination function via the RFWD3[C315A] mutant was the only condition that decreased absolute C•G to G•C editing efficiencies, suggesting this may have resulted in preferential conversion of the intermediate toward other outcomes.

In an effort to explore strategies that specifically enhance the C•G to G•C outcome, we co-transfected wild-type HEK293T cells with plasmids encoding evo-BE4(ΔUGI), sgRNA targeting either the *RNF2* or *HEK2.1* protospacers, and one of the three RFWD3 variants (Supplementary Fig. 14g). We found that none of these overexpression strategies caused any significant changes in relative C•G to G•C editing efficiencies and in fact consistently caused a reduction in absolute editing efficiencies. This may be due to the large size of this additional gene (the protein product is 774 amino acids) causing a reduction in expression or cell fitness.

## Competition and cooperation of multiple genes shape mixed editing outcomes

We additionally examined the effects of *POLD1*, *ERCC4*, or *LIG3* knockdown on editing efficiencies by rA1-SaBE4(ΔUGI), rA1-SaBE4, and rA1-SaBE1 at the *HEK3* and *RNF2* sites in K562 cells. These data, along with that from *UNG*, *MSH2*, *MSH6*, and *RFWD3* are shown in Fig. 6a, b for all editing outcomes, including mixed C•G to T•A and C•G to G•C dual edits. We observed high toxicity following transfection of cells in which *LIG3* or *POLD1* was knocked down, presumably due to the essential roles these genes play in maintaining genomic stability in the face of DNA damage. Consequently, we observed minimal impact of *POLD1* knockdown on editing outcomes, except for reductions in the frequence of dual editing

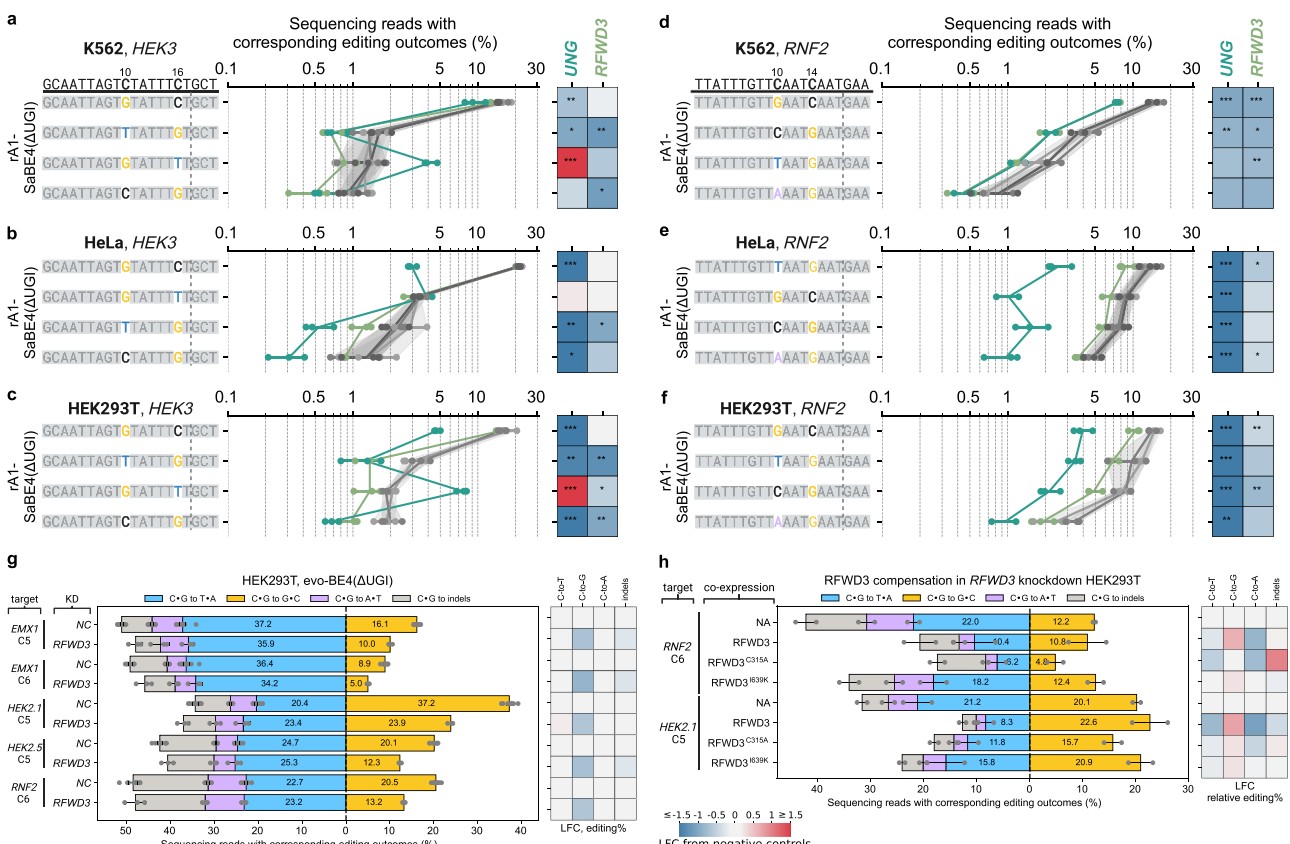

**Fig. 5 | Impacts of *RFWD3* and *UNG* knockdown on C•G to G•C outcomes in K562, HeLa, and HEK293T cells.** Cells were treated as described in Supplementary Fig. 12a (K562), 13a-b (HeLa), or 14a-c (HEK293T). Shown are effects of indicated gene knockdown on the top four outcomes with C•G to G•C conversions at the *HEK3* site in K562 (**a**), HeLa (**b**) and HEK293T (**c**), and at the *RNF2* site in K562 (**d**), HeLa (**e**) and HEK293T (**f**). Protospacer sequences are shown at the top. Vertical dashed lines mark the Cas9-induced nick positions. Tract plots display absolute frequencies of editing outcomes upon specific gene knockdown (color-coded by gene as indicated). Three negative control samples without knockdown are shown in gray colors, with shaded areas representing ±2 STDs of mean. Dots show individual replicates, and tract lines show averages of *n* = 2 (**d**) and *n* = 3 (all the rest) biological replicates. Horizontal lines show ranges of the replicates for corresponding outcomes. Heatmap shows log₂ (fold change) (LFC) in frequencies of editing outcomes relative to the mean of negative controls. *: 1.5 <adjusted z-score

<2. **: 2<adjusted z-score <3. ***: adjusted z-score > 3. **g** Shown are editing efficiencies of all outcomes at five endogenous target Cs in negative control (NC) and *RFWD3* knockdown cell lines. The bars and the error bars represent the mean and propagation of uncertainty of standard deviation (STD) obtained from *n* = 3 biological replicates. Heatmap shows log₂ (fold change) (LFC) in frequencies of editing outcomes relative to the corresponding means of negative controls. **h** Shown are editing efficiencies of all outcomes at two endogenous targets upon compensation of indicated RFWD3 mutants in *RFWD3* knockdown HEK293T cells. Negative control (NA) cells were treated identically without compensation. The bars represent the mean obtained from *n* = 2 biological replicates. Lines show ranges of the replicates for corresponding outcomes. Heatmap shows log₂ (fold change) (LFC) in relative percentages of editing outcomes relative to the corresponding means of negative controls. Source data are provided as a Source Data file.

outcomes containing either two C•G to T•A edits or one C•G to T•A edit combined with one C•G to G•C edit when using rA1-SaBE4(-ΔUGI) at the *RNF2* site. Furthermore, *LIG3* knockdown did not cause any statistically significant changes in editing outcomes when base editors were transfected, in contrast to the integrated editing system. *ERCC4* knockdown increased the frequency of certain dual editing outcomes when using rA1-SaBE4(ΔUGI) at both sites. Additionally, we observed *ERCC4* knockdown to consistently increase all editing outcomes when using rA1-SaBE1, indicating a potential additional role for this gene in facilitating mutational conversions in the absence of the Cas9-induced nick.

When analyzing mixed editing outcomes, we observed that their editing frequencies were often influenced by multiple repair factors. For example, at the *HEK3* site, the dual T10/G16 outcome (observed only with rA1-SaBE4(ΔUGI)) was decreased by knockdown of *UNG* or *RFWD3*, but increased by knockdown of *ERCC4*. At the *RNF2* site, the T10/G14 edit (observed only with rA1-SaBE4(ΔUGI)) was decreased by knockdown of *RFWD3* but increased by *ERCC4* knockdown. Meanwhile, the dual T10/T14 edit was decreased by knockdown of *MSH6* but

increased by *UNG* knockdown. When UNG was inhibited by UGI (when using the rA1-SaBE4 construct), MutSα emerged as the primary factor influencing C•G to T•A conversions. In the absence of the Cas9-induced nick (when using the rA1-SaBE1 construct), *UNG* knockdown consistently increased C•G to T•A outcomes, while *ERCC4* knockdown increased all outcomes.

We then repeated our HeLa cell knockdown experiments with *LIG3* and *ERCC4* (Supplementary Fig. 15). In HeLa cells, UNG exhibited a significantly more dominant role than in K562 cells, as reflected by the greater fold changes in outcomes following *UNG* knockdown. Knockdown of *MSH2* did not significantly impact C•G to T•A outcomes at either locus when using rA1-SaBE4(ΔUGI), but was the only gene to demonstrate a significant impact on editing outcomes when UNG is inhibited by rA1-SaBE4 (Supplementary Fig. 15a). *RFWD3* knockdown resulted in similar effects in both HeLa and K562 cells. However, *ERCC4* knockdown did not produce the same significant increase in many of the editing outcomes in HeLa cells as was observed in K562 cells, potentially due to the near saturated editing efficiencies in HeLa cells (Supplementary Fig. 13d, e).

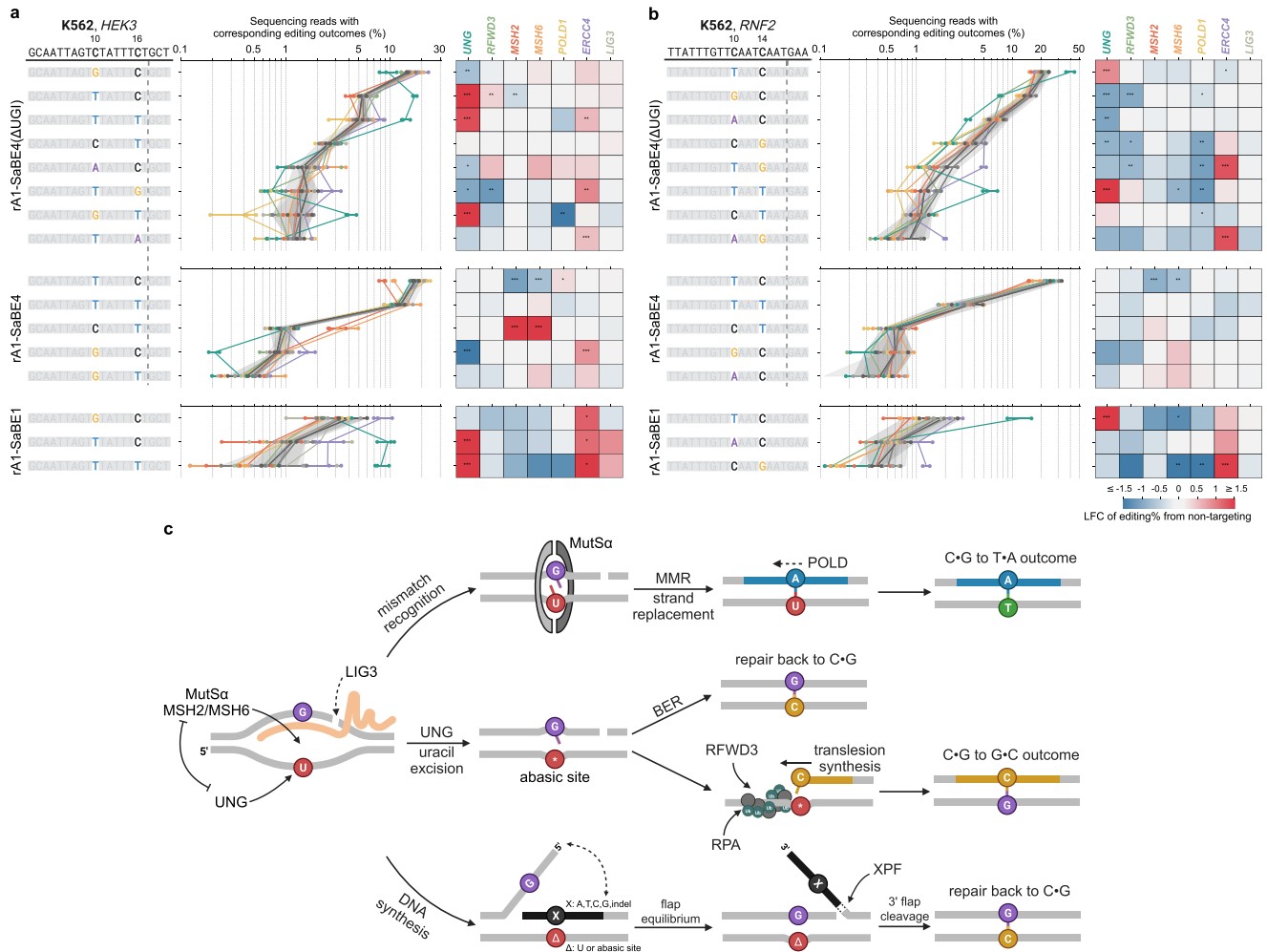

**Fig. 6 | Competition and cooperation of multiple genes shape base editing outcomes.** Effects of indicated gene knockdown on editing outcomes at the *HEK3* (**a**) and *RNF2* (**b**) sites in K562 cells. Protospacer sequences are shown at the top, and base editors used are labeled on the left. The top eight, five, and three editing outcomes for rA1-SaBE4(ΔUGI), rA1-SaBE4, and rA1-SaBE1 [rA1-dSaCas9(ΔUGI)] are listed, respectively. Tract plots display absolute frequencies of editing outcomes upon specific gene knockdown (color-coded by gene as indicated). Three negative control samples without knockdown are shown in gray colors, with shaded areas representing ±2 STDs of mean. Dots show individual replicates, and tract lines show averages of $n = 2$ (**b**: NCs, *UNG*, *MSH2*, *LIG3*, *RFWD3*, *ERCC4*) and $n = 3$ (all the rest)

biological replicates. Horizontal lines show ranges of the replicates for corresponding outcomes. Heatmap shows $\log_2$ (fold change) (LFC) in frequencies of editing outcomes relative to the corresponding means of negative controls. *: 1.5 <adjusted z-score <2. **: 2<adjusted z-score <3. ***: adjusted z-score > 3. (**c**) Proposed mechanisms of cytosine base editing. Competition and cooperation among indicated genes influence base editing outcomes. Dash lines indicate the effects of those genes were only observed in the integrated rA1-SaBE4(ΔUGI) system where only one target C was edited. Created in BioRender. Gu, S. (2025) https://BioRender. com/untq2zw. Source data are provided as a Source Data file.

## Discussion

Here, we have identified through CRISPRi screens several genes that regulate cytosine base editing outcomes. Our data suggest that the unique mutagenic intermediate of cytosine base editing is recognized and processed by multiple genes, and the genetic landscapes governing different outcomes vary depending on the targeting sequences, editing kinetics, and possibly cell type. Notably, our data suggest that intermediates involving multiple uracils are processed via a different mechanism than single uracil intermediates.

A mechanism that is consistent with our observations is one in which UNG and the MutSα complex (the MSH2/MSH6 heterodimer) compete to recognize the U•G intermediate, with UNG generally dominating this competition. While recognition by UNG facilitates C•G to non-T•A outcomes, recognition by MutSα promotes the C•G to T•A outcome (Fig. 6c)[32,33]. Natively, MutSα recognition initiates downstream MMR processes resulting in replacement of the DNA strand that is nicked by MutLα[34,35]. In cytosine base editing, this strand discrimination signal is potentially provided by the Cas9-induced nick on

the non-edited strand. Hence, replacement of the non-edited strand would facilitate the C•G to T•A outcome. Under conditions where only one target C is edited, we observed LIG3 to inhibit strand replacement, potentially by sealing the Cas9-induced nick, and POLD1 to facilitate strand replacement, likely through its role in DNA synthesis.

Excision of uracil by UNG generates an abasic site, which can be processed by downstream BER to achieve the fully repaired outcome (Fig. 6c)[4]. Alternatively, DNA synthesis may occur before BER-mediated processing of the abasic site, which can lead to the activation of translesion synthesis due to stalling of replicative polymerases at the abasic site[36,37]. Our data are consistent with a mechanism in which RFWD3 plays an important role in facilitating the translesion synthesis pathway that specifically leads to the C•G to G•C outcome. When replication stalls, ssDNA becomes exposed and is subsequently bound by RPA. RFWD3 is recruited to the site of stalled replication through its interaction with RPA and then initiates a series of ubiquitylation reactions of RPA and nearby proteins. This process eventually leads to mono-ubiquitylation of PCNA, which promotes the

recruitment of Y-family error-prone polymerases to carry out DNA synthesis over the replication-blocking lesion[12,13,31]. We show both the RPA-binding and ubiquitylation activities of RFWD3 are essential for its role in facilitating the C•G to G•C outcome. However, the specific error-prone polymerase responsible for incorporating a deoxycytidine across the abasic site in cytosine base editing remains unclear, as knockdown of either *REV1* or *POLH* only marginally reduced C•G to G•C editing efficiencies (Supplementary Fig. 8). This is consistent with other studies that attempted to identify the error-prone polymerase responsible for the C•G to G•C outcome[7,8]. Future studies that perturb multiple genes simultaneously may be required to provide further insights into this process[37,38].

Studies have shown that XPF (encoded by *ERCC4*) is a structure-specific endonuclease that specifically cleaves at the junction of double- and single-stranded DNA of 3' flap structures (Fig. 6c)[26]. In the case of cytosine base editing, a 3' flap can arise from displacement of the newly synthesized strand by the old non-edited strand via flap equilibrium. Cleavage of the newly synthesized strand abolishes mutation formation, which is consistent with the increase in certain editing outcomes at the two endogenous sites upon *ERCC4* knockdown (Fig. 6a, b).

While DNA mismatch repair has been hypothesized to participate in base editing since its invention, this has not been experimentally validated previously[3]. Furthermore, we identified additional DNA repair genes that were previously not known to resolve the cytosine base editing intermediate, and discovered the role of RFWD3 in specifically mediating the C•G to G•C outcome. Overall, this study sheds light on several key aspects of the cellular processing of the unique DNA damage intermediate leveraged in cytosine base editing.

## Methods
### Plasmid design and cloning
A full list of all plasmids used in this study is included in Supplementary Data 4, 5. The pMH0001 (dCas9-BFP-KRAB) was obtained from Addgene (#85969). All cloning primers were obtained from Integrated DNA Technologies (IDT). The dCas9-KRAB-MeCP2-P2A-mCherry lentiviral vector (pSG231) was derived from dCas9-KRAB-MeCP2 (Addgene, 110821). First, an intermediate plasmid was generated in which a P2A-mCherry cassette was added to the end of the dCas9-KRAB-MeCP2 insert. USER cloning was then used to construct pSG231, in which the dCas9-BFP-KRAB insert in pMH0001 was replaced with dCas9-KRAB-MeCP2-P2A-mCherry[39].

All rA1-SaBE4(ΔUGI) plasmids were derived from SaBE4 (Addgene, #100805). The UGI portion of SaBE4 was replaced with a P2A-miRFP670 cassette to construct pSG199 (the rA1-SaBE4(ΔUGI)) variant with rAPOBEC1). eA3A and RrA3F templates were obtained from p2T-eA3Amax (Addgene, 152999) and pCMV-BE4-RrA3F (Addgene, 138340), respectively. The rA1 portion of pSG199 was replaced with eA3A or RrA3F via USER cloning to construct pSG200 and pSG201, respectively. The backbone of the rA1-SaBE4(ΔUGI) PiggyBac transposon plasmid, pSG236, was adopted from PB-TRE-dCas9-VPR (Addgene, #63800). rA1-SaBE4(ΔUGI) was assembled into the PiggyBac backbone with USER cloning to produce pSG236.

The backbone of both reporter vectors was adopted from pU6-sgRNA EF1α-puro-T2A-BFP (Addgene, #60955). A hU6-2xBsmBI-Sa-sgRNA cassette was inserted in between the pU6-sgRNA cassette and EF1α promotor, and the wild-type BFP was replaced with the BFP reporter variant by USER cloning to construct pSG297. To construct the lentiviral C•G to T•A reporter vector pSG298, the targeting spacer sequence was introduced into the Sa-sgRNA expression cassette via golden gate cloning using BsmBI-v2 (R0739L, NEB)[40]. To construct the lentiviral C•G to G•C reporter vector pSG299, the entire EF1α-puro-T2A-BFP cassette in pSG297 was replaced with the C•G to G•C reporter (dGFP-2bp-P2A-mCherry) and EF1α-puro cassettes, followed by introduction of the targeting spacer sequence to the Sa-sgRNA expression

cassette by BsmBI golden gate cloning. For validation at BFP/dGFP reporter sites, individual CRISPRi Sp-sgRNAs nominated from screens were cloned into the corresponding reporter vectors (pSG298 or pSG299) via BstXI/BlpI (BstXI: R0113L, BlpI: R0585L, NEB) restriction enzyme cloning to produce single CRISPRi Sp-sgRNA vectors (Supplementary Data 5). For validation at endogenous sites, CRISPRi Sp-sgRNAs were first cloned into pSG297 by BstXI/BlpI restriction enzyme cloning, followed by incorporation of the spacer sequences of Sa-sgRNA targeting endogenous sites by BsmBI golden gate assembly (Supplementary Data 5). For cDNA re-expression experiments, the BFP portion of these plasmids was replaced with the corresponding cDNA sequence (Supplementary Data 6). The cDNA of *UNG* was obtained from reverse transcription of mRNA in HEK293T. cDNA sequences of *RFWD3*, *ERCC4* and *MSH2* were ordered from GenScript (Supplementary Data 6).

Dual sgRNA vectors used in generating HeLa and HEK293T stable knockdown cell lines were cloned according to an established protocol[41]. Briefly, oligos were designed containing two spacer sequences (ps1 and ps2) separated by 2xBsmBI recognition sites with BstXI and BlpI sites at the 5'- and 3'-ends, respectively (Supplementary Data 6). Spacer sequences for each gene are listed in Supplementary Data 5. Oligos were ordered (IDT) and cloned into the pSG942 lentiviral backbone (mU6-BstXI/BlpI-UCOE-EF1α-puro-T2A-EGFP) via BstXI/BlpI restriction enzyme cloning. Next, the "drop-in" sequence containing the Sp-sgRNA backbone and hU6 promoter (Supplementary Data 5) was inserted in between the two spacer sequences by BsmBI golden gate cloning to produce the full dual sgRNA expression vector (Supplementary Data 5).

### Cell culture and cell line construction
All cell lines were cultured at 37 °C with 5% $CO_2$ and 95% relative humidity. HEK293T (ATCC, CRL-1573) and HeLa cells (ATCC, CRL-3216) were cultured in Dulbecco's Modified Eagle Medium (DMEM) supplemented with GlutaMAX (10566-024, Thermo Fisher) and 10% (v/v) fetal bovine serum (10437-028, Thermo Fisher). K562 cells (ATCC, CCL-243) were grown in Roswell Park Memorial Institute (RPMI) 1640 Medium (11875-119, Thermo Fisher) supplemented with 10% (v/v) fetal bovine serum and 1% penicillin/streptomycin (15140122, Gibco). All cell lines were authenticated by the supplier (ATCC) with short tandem repeat (STR) profiling and tested negative for mycoplasma with Mycostrip (InvivoGen, rep-mys-10) every two months.

To construct the CRISPRi-expressing cell line for the C•G to T•A screen, wild-type K562 cells were transduced with the dCas9-KRAB-MeCP2-P2A-mCherry lentiviral vector (pSG231) described above. To construct the CRISPRi-expressing cell line for the C•G to G•C screen, wild-type K562 cells were transduced with the dCas9-BFP-KRAB lentiviral vector (pMH0001). These transductions were performed as described below. Pure polyclonal populations of each CRISPRi-expressing cell line were then generated by sorting the transduced cells on a BD FACS Aria II for the top 50% of mCherry or BFP signal, respectively.

Doxycycline-inducible rA1-SaBE4(ΔUGI) expression and hygromycin resistance cassettes (pSG236) were then incorporated into each CRISPRi-expressing cell line via PiggyBac transposition. Briefly, 100 ng transposon vector (pSG236) and 20 ng PiggyBac transposase vector (PB210PA-1, SBI) were co-transfected into 20,000 CRISPRi-expressing K562 cells using Lipofectamine 2000 (11668027, Invitrogen). Two days after transfection, cells with stable integration were selected by a 2-week, 200 µg/ml hygromycin treatment.

To construct CRISPRi HeLa cells, wild-type HeLa cells were transduced with the dCas9-BFP-KRAB lentiviral vector (pMH0001) at an infection rate of approximately 30%. Pure polyclonal population of CRISPRi-expressing cells were collected by sorting the transduced cells on a BD FACS Aria II for the top 50% of BFP signal. A second sort was

performed after a 7-day expansion of the collected cells from the first sort to ensure a > 90% BFP+ population.

## Evaluation of different deaminases and validation of reporter constructs

HEK293T cells were seeded at 100,000 cells/well in 250 μL media in a 48-well plate and transfected four hours later with a 25 μL DNA/Lipofectamine 2000 mixture. This mixture was prepared by diluting 950 ng of a DNA mixture (described below) to 12.5 μL total volume with Opti-MEM reduced-serum medium (31985-070, Thermo Fisher) and adding this to a mixture of 1.5 μL Lipofectamine 2000 and 11 μL Opti-MEM. The DNA mixture was comprised of 500 ng of a SaCas9-BE4(ΔUGI) vector with one of the rAPOBEC1 (pSG199), eA3A (pSG200), or RrA3F (pSG201) deaminases, 300 ng of either the C•G to T•A (pSG139) or C•G to G•C transient reporter vector (pSG154), and 150 ng of a sgRNA vector (pSG157 or pSG158 for the C•G to T•A reporter and C•G to G•C reporter, respectively). 72 h post-transfection, cells were analyzed by flow cytometry using a BD LSRFortessaX20 for either BFP to GFP conversion (for the C•G to T•A reporter) or GFP turn-on (for the C•G to G•C reporter). A total of 10,000 events were analyzed in Flowjo v10.9 for each sample to calculate conversion or turn-on rates.

## CRISPRi library design and construction

The 12,318 sgRNA CRISPRi library was adapted from a previously published library (Richardson et al.)[42]. In Richardson's library, a set of 2015 genes enriched for those involved in DNA metabolic processes and the cellular response to DNA damage was curated (Supplementary Data 1). The targeting sgRNAs and 2243 non-targeting control sgRNAs from this library originated from hCRISPRi-v2 (Horlbeck et al., 2016)[43]. The CRISPRi sgRNA library was constructed using the protocol available at https://weissman.wi.mit.edu/crispr/. Briefly, an oligonucleotide pool containing these spacer sequences was synthesized by Agilent. Primer binding sites, BstXI and BlpI recognition sites, and the appropriate overhang sequence for cloning into the reporter vectors, were appended to the 5' and 3' ends of each spacer sequence (Supplementary Data 6). The oligonucleotide pool was amplified by PCR using Phusion DNA Green High-Fidelity Polymerase (F534L, Thermo Fisher), purified with the MinElute PCR purification kit (28004, QIAGEN), and digested with BstXI and BlpI. The digested oligonucleotide library was then isolated by polyacrylamide gel electrophoresis followed by ethanol precipitation. The C•G to T•A (pSG298) and C•G to G•C reporter vectors (pSG299) were also digested with BstXI and BlpI and purified by agarose gel electrophoresis and the QIAquick gel extraction kit (28704). The digested spacer library was then ligated into both digested reporter vectors by incubating the two DNA pieces with an insert to backbone ratio of 2:1 at 16 °C for 16 h with T4 DNA ligase (M0202, New England Biolabs). 50 ng of the ligation mixture was then electroporated into 50 μl of NEB 10-beta cells (C3019, New England Biolabs). Greater than 300x library coverage was achieved at transformation for both the C•G to T•A and C•G to G•C library by performing at least 5 electroporations. Cells were grown in 2XYT media (200 mL) supplemented with 100 μg/ml carbenicillin at 37 °C for 16 h, after which the library-containing plasmids were purified using the Zymo-PURE II plasmid midiprep kit (D4201, Zymo).

## Lentivirus production

For small-scale viral production when creating cell lines, $3 \times 10^5$ HEK293T cells per well were seeded in 2 mL of media in 6-well plates and transfected 24 h later with a 300 μL DNA/TransIT-LT1 (MIR2306, Mirus) solution. This mixture was prepared by diluting a DNA mixture consisting of 165 ng pMD2.G (12259, Addgene), 1350 ng pCMV-delta-R8.2 (12263, Addgene), 100 ng pMax_GFP, and 1500 ng transfer plasmid to 292.5 μL total volume with Opti-MEM. Then, 7.5 μL TransIT-LT1 was added to this mixture. These DNA/TransIT-LT1 solutions were incubated at room temperature for 15 min and added dropwise into

each well. Viral supernatant was harvested 72 h after transfection and filtered through a 0.45 μm PES syringe filter (725-2545, Thermo Scientific).

To perform large-scale viral productions for the pooled CRISPRi screens, $3 \times 10^6$ HEK293T cells per 15 cm² dish were seeded in 30 mL of medium and then transfected 24 h later with a 1 mL DNA/TransIT-LT1 solution. This mixture was prepared by diluting a DNA mixture consisting of 2 μg pMD2.G, 16 μg pCMV-delta-R8.2, 2 μg pMax_GFP, and 16 μg transfer plasmid (containing the CRISPRi sgRNA library) to 1000 μL total volume with Opti-MEM. Then, 48 μL TransIT-LT1 was added to this mixture. These DNA/TransIT-LT1 solutions were incubated at room temperature for 15 min and added dropwise into each dish. 24 h post-transfection, 60 μL of viral boost (VB100, Alstembio) was added to each dish according to the manufacturer's instructions. 72 h post-transfection, the viral supernatant of all dishes with the same library was pooled and filtered through a 0.45 μm PES vacuum filter.

## Lentiviral titering

To determine the lentiviral titer for transductions of pooled CRISPRi sgRNA libraries, $12 \times 10^6$ cells of the appropriate K562 cell line were plated in 2 mL media per well in 6 well plates and transduced with 0, 1 ml, 3 ml, or 6 ml virus supplemented with 8 μg/ml polybrene (TR-1003-G, Sigma). The plates were centrifuged at 1000× g for 2 h at 33 °C. Virus-containing media was then aspirated, and cells were resuspended at $0.5 \times 10^6$ cells/ml in fresh media. Cells infected with the C•G to T•A reporter vector were analyzed for BFP fluorescence 2 days after transduction using flow cytometry on a BD LSRFortessa X20. Cells infected with the C•G to G•C reporter vector were split equally into twin wells in a 6 well-plate and 2.5 μg/ml puromycin was applied to one of the two wells for 3 days. On day 3, the number of cells in both wells were counted to calculate viability. A viral dose resulting in around 10% infection rate was then chosen to perform the pooled screens.

## CRISPRi pooled screens

For both the C•G to T•A and C•G to G•C screens, cells were transduced in 2 biological replicates with their corresponding lentiviral library. For both screen, transductions were performed at a low multiplicity of infection (MOI < 0.1), using $2 \times 10^8$ cells to ensure a representation of at least 500× coverage for each sgRNA after transduction. $1.2 \times 10^7$ cells were plated per well in 6-well plates to give a density of $2 \times 10^6$ cells/ml with virus-containing media and 8 μg/ml polybrene. Plates were centrifuged at 1000× g for 2 h at 33 °C. After centrifugation, cells were pooled, spun down again at 300× g for 10 minutes and resuspended in fresh RPMI media to a density of $0.5 \times 10^6$ cells/ml. Two days post-transduction, 2.5 μg/ml puromycin was added to the media, and cells were maintained for another 3 days. The day on which puromycin selection was concluded was designated as day 0 for the screens. To perform the screens, on day 0 cells were resuspended into fresh RPMI media to a density of $5 \times 10^5$ cells/ml and then split into an experimental arm and a control arm, each containing at least $1.5 \times 10^7$ cells (>1000× sgRNA coverage). An additional $1.5 \times 10^7$ cells were harvested for genomic DNA extraction using the Nucleospin Blood L kit (740954.20, Macherey-Nagel) to serve as a control on day 0. The cells in the experimental arm were treated with 100 ng/ml doxycycline (MP219504410, MP Biomedicals), while cells in the control arm continued to be cultured without doxycycline. Both arms were passaged every two days while maintaining at least 1000× sgRNA coverage. On day 13, $3.0 \times 10^7$ cells from the experimental arm (representing >2000x sgRNA coverage) were sorted into populations of interest (GFP+ population for the C•G to T•A screen, GFP + / mCherry- and GFP-/mCherry+ populations for the C•G to G•C screen) on a BD FAC-SAria II or Fusion. An additional $1.5 \times 10^7$ cells from the experimental arm and $1.5 \times 10^7$ cells (representing >1000x sgRNA coverage each) from the control arm were also collected on day 13 to serve as unsorted

controls. The genomic DNA from all collected populations was then extracted using the Nucleospin blood L kit.

## CRISPRi pooled screen library sequencing

To quantify the base editing efficiencies of the screens, 5% of the genomic DNA collected from each population was used to amplify the *BFP* (C•G to T•A screen) or *dGFP* (C•G to G•C screen) locus. The remainder of the genomic DNA from each population was used in entirety to amplify the CRISPRi sgRNA region. Amplification primer sequences are provided in Supplementary Data 6. Round 1 PCRs were set up using Phusion DNA Green High-Fidelity Polymerase (F534, ThermoFisher Scientific) with 5 µg of genomic DNA per 100ul reaction volume. These reactions also contained GC buffer, 3% DMSO and 1 µM each primer. The thermocycler was set with the following program: 1 cycle of 98 °C for 2 min, 23 to 26 cycles of 98 °C for 30 s, 60 °C for 30 s and 72 °C for 20 s, and 1 cycle of 72 °C for 10 min followed by holding at 4 °C. PCRs from the same population were pooled and cleaned up using the QIAquick PCR purification kit (QIAGEN), and target bands were further isolated by gel extraction. Round 2 PCRs were conducted using 10 ng of purified DNA from round 1 PCRs per 100ul reaction volume to append unique sequencing barcodes to samples from different populations. Round 2 PCRs contained HF buffer and 0.125 µM of each primer, and were run on a thermocycler with the following program: 1 cycle of 98 °C for 30 s, 10 cycles of 98 °C for 15 s, 65 °C for 30 s and 72 °C for 15 s, and 1 cycle of 72 °C for 10 min followed by holding at 4 °C. Samples were pooled, purified by gel extraction, and then quantified by a Qubit using the dsDNA HS assay kit (Q32854, Thermo Fisher). Sequencing was performed on an Illumina MiniSeq using the high output reagent kit (FC-420-1002) with 10% spike-in of PhiX. Enough sequencing reads were assigned to each population to ensure at least 200× reads per sgRNA.

## CRISPRi screen data analysis

CRISPRi pooled screen data analysis was performed using MAGeCK RRA (v0.5.9.4)[44,45]. Briefly, reads from different samples were median-normalized, and zero-count sgRNAs were removed. Gene phenotype was defined as the median $\log_2$ fold change of the 5 target CRISPRi sgRNAs, and *P* values were computed based on the negative binomial model using a modified robust ranking aggregation algorithm. For gene ontology enrichment analysis, genes with $-\log_2(P)$ values greater than 1.5 were searched for enriched gene ontology biological processes (GOBP) using MAGeCKFlute.

## RT-qPCR

For RT-qPCT analyses, $1 \times 10^5$ to $2 \times 10^5$ cells were collected, and their RNA was extracted with the Quick-RNA Miniprep Kit (Zymo, R1054). cDNA was produced from 300 ng of purified RNA using the SuperScript First-Strand Synthesis System for RT-qPCR (Invitrogen, 11904018). qPCR reactions were conducted with PowerUp SYBR Green Master Mix (Applied Biosystem, A25742) in a total volume of 10 µL with 500 nM primers following the manufacturer's protocol. Gene expression levels were normalized to *ACT1B* using the $-2^{\Delta\Delta C_t}$ method[46].

## Arrayed validation experiments in K562 cells

To generate stable knockdown and cDNA-expressing cell lines for each gene of interest, we individually transduced the single CRISPRi Sp-sgRNA vectors listed in Supplementary Data 5 into the CRISPRi (dCas9-BFP-KRAB) and dox-inducible rA1-SaBE4(ΔUGI) K562 cell line at a MOI < 0.3. This cell line was chosen for validation (rather than the dCas9-KRAB-MeCP2-P2A-mCherry-expressing cell line) as it demonstrated better knockdown efficiency during the screens based on essential gene depletion. Transduced cells were selected with a 3-day, 2.5 µg/ml puromycin treatment.

For dox-induction experiment, stable knockdown K562 cells were treated with 100 ng/ml doxycycline for seven days. -100,000 cells were collected on day 7 and lysed using cell lysis buffer (10 mM Tris, 0.05% SDS and 1.25 U/ml proteinase K).

For transfection experiments, 200,000 stable knockdown K562 cells were nucleofected with 800 ng of base editor plasmid, encoding either rA1-SaBE4(ΔUGI)-P2A-mCherry, rA1-SaBE4-P2A-mCherry or rA1-SaBE1-P2A-mCherry, and 200 ng of sgRNA plasmid using the SF Cell Line 4D-Nucleofector X Kit S (Lonza, V4XC−2032) and pulse program FF-120, according to the manufacturer's protocol. After nucleofection, cells were resuspended in RPMI and cultured in 12-well plates. Three days post transfection, transfected cells were collected by sorting the mCherry+ population on a Bio-Rad S3e cell sorter using Biorad ProSort software v1.6 and lysed using cell lysis buffer at a density of 1000 cells/µl.

## Arrayed validation experiments in HeLa cells

To establish HeLa cell lines with stable knockdown of genes of interest, CRISPRi HeLa cells were transduced with the dual sgRNA vectors listed in Supplementary Data 5 at a MOI < 0.3. Two days post transduction, puromycin was applied at 1.5 µg/ml for 3 days to select for transduced cells. 350,000 cells from each HeLa stable knockdown cell line were seeded in 2 mL of media in 6-well plates and transfected 24 h later with 3 µg base editor plasmid, encoding either rA1-SaBE4(ΔUGI)-P2A-mCherry or rA1-SaBE4-P2A-mCherry, and 750 ng sgRNA plasmid. Three days post transfection, transfected cells were collected by sorting the mCherry+ population on a Bio-Rad S3e cell sorter using Biorad ProSort software v1.6 and lysed using cell lysis buffer at a density of 1000 cells/µl.

## Arrayed validation experiments in HEK293T cells

To establish HEK293T cell lines with stable knockdown of genes of interest, 40,000 wildtype HEK293T cells were seeded in 250 µL of media in 48-well plates and transfected 24 h later with 180 ng CRISPRoff_v2.1 (Addgene, #167981) and 90 ng dual sgRNA vector[47]. One day post transfection, puromycin was applied at 3 µg/ml for 3 days to select for transfected cells. All cell lines were then passaged for 7 days to allow for the degradation of the CRISPRoff and dual sgRNA vectors.

For transfection, 40,000 cells from each HEK293T stable knockdown cell line were seeded in 250 µL of media in 48-well plates and transfected 24 h later with 400 ng base editor plasmid, encoding either rA1-SaBE4(ΔUGI)-P2A-GFP, rA1-SaBE4-P2A-GFP, or evo-BE4(ΔUGI)-P2A-mCherry, and 100 ng sgRNA vector. Three days post transfection, transfected cells were collected by sorting the GFP+ or mCherry+ population on a Bio-Rad S3e cell sorter and lysed using cell lysis buffer at a density of 1000 cells/µl.

## NGS sample preparation and data analysis for arrayed validation experiments

Editing efficiencies of the target regions were quantified by targeted amplicon NGS. 0.5 to 2.5 µL of cell lysis was added in a 12.5 µL PCR reaction with 0.1 to 0.125 µM primers for round 1 PCR and amplified for 25 cycles. After confirmation on a 2% agarose gel, round 2 PCR was performed to barcode samples with 8-12 cycles. Samples were pooled, purified by gel extraction, and then quantified by Qubit with the dsDNA HS assay kit. Sequencing was performed on an Illumina MiniSeq (2 × 151 paired end reads) per the manufacturer's instructions.

Targeted amplicon NGS data were first processed using Illumina Local Run Manager Generate FASTQ analysis module (v2.0) on MiniSeq control software (v2.2.1) to demultiplex and trim Fastq files. Data were then analyzed using CRISPResso2 (version 2.0.20b) with default settings and the following options: --base_editor_output, -wc −10, -w 10, --plot_window_size 10[48]. Alleles frequencies files were extracted for further analysis to generate tract plots.

Z-scores were computed as follow:

$$z = \frac{x - \mu}{\sigma} \qquad (1)$$

where $\mu$ is the mean of negative controls, and $\sigma$ is the standard deviation of negative control distribution.

We computed adjusted z-scores for samples with multiple replicates:

$$\text{adjusted z} - \text{score} = |\text{mean}(z)| - \sigma(z) \qquad (2)$$

where |mean(z)| is the absolute value of the average z-score from multiple replicates, and $\sigma(z)$ is the standard deviation of the z-score distribution.

## Data visualization

All processed data were visualized using Python 3.9.5, Pandas v1.4.2, Matplotlib v3.4.3, and/or Seaborn v0.11.2.

## Reporting summary

Further information on research design is available in the Nature Portfolio Reporting Summary linked to this article.

## Data availability

The next-generation sequencing data generated in this study have been deposited in the NCBI Sequencing Read Archive (SRA) under project number PRJNA1081436. Plasmids generated in this study are available at Addgene (Addgene Plasmid #s 239447, 239448, 129450, 239461, 239462, and 239463). Source data are provided with this paper.

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

## Acknowledgements

This research was supported by the University of California, San Diego and the National Science Foundation through grant no. MCB-2048207 (to A.C.K.). Z.B. was supported by the Research Corporation for Science Advancement through Cottrell fellowship no. 27975 (to Z.B.). R.A.A. was supported by the Chemistry-Biology Interfaces Training Grant, NIH Grant T32GM146648. We are grateful to Cody Fine, Mateo Espinoza and Mitra Banihassan (UCSD) for technical assistance with flow cytometry experiments. This work was additionally made possible by the UC San Diego Stem Cell Program and a CIRM Major Facilities grant (FA1-00607) to the Sanford Consortium for Regenerative Medicine. This publication includes data generated at the UCSD Human Embryonic Stem Cell Core Facility, using the BD FACSAria II and Fusion.

## Author contributions

S.G. contributed to conceptualization of the research project, experimental design, data curation, data analysis, and visualization, and writing of the manuscript. Z.B. contributed to conceptualization of the research project, funding acquisition, experimental design, data curation, data analysis, and writing of the manuscript. S.G. and Z.B. authors contributed equally of this manuscript. R.A.A. contributed to data curation. H.Y.A.S. contributed to data curation. Q.T.C. contributed to conceptualization of the research project and data curation. A.C.K. contributed to conceptualization of the research project, experimental design, data analysis, supervision of the work, writing of the manuscript, and acquisition of the funding. All authors contributed to editing of the manuscript.

## Competing interests

A.C.K. is a member of the SAB of Pairwise Plants, is an equity holder for Pairwise Plants and Beam Therapeutics, and receives royalties from Pairwise Plants, Beam Therapeutics, and Editas Medicine via patents licensed from Harvard University. A.C.K.'s interests have been reviewed and approved by the University of California, San Diego in accordance with its conflict of interest policies. All other authors declare no competing interests.
