## [Transparent Peer Review file · Nature Communications]

Elucidating the genetic mechanisms governing cytosine base editing outcomes through CRISPRi screens

Corresponding Author: Professor Alexis Komor

Version 0:

Reviewer comments:

Reviewer #1

(Remarks to the Author)

The manuscript by Gu et al aims to elucidate the molecular mechanisms underlying cytosine base editing (CBE) through CRISPRi screening. The author applied two distinct GFP reporter systems in conjunction with CRISPRi to identify factors involved in the process of C•G to T•A or C•G to G•C base editing. They screened out MMR factors promoting C•G to T•A, and TLS factors promoting C•G to G•C, while NER factors inhibiting C•G to G•C. Then they verified the top hits in HEK293T and HeLa cell lines. In general, this study corroborates most findings from previous reports (doi: 10.1038/s41556-023-01342-2; doi: 10.1038/s41587-021-00938-z). Its novelty within the field appears limited. Furthermore, the verifications of their screening results are inconclusive, and it remains unclear how these genes specifically guide the optimization of the base editors.

Major points:

1. The authors spent too much space on the introduction of the CRISPRi system and quality control, which could be shown with bar graphs instead of long text. And most of the analysis process could be excluded from the result context, and move to the Method part.
2. The sequencing depth and real read counts in each editing outcomes should be included to better convince the readers.
3. In the section "Identification of genes that are synthetic lethal to CBE expression and/or activity", the authors use "bulk, +dox samples" vs. "bulk, -dox samples" to screen out genes that are lethal to CBE function. Did they exclude genes that are essential for cell survival? To define essential genes, compare of "bulk, -dox samples" to "sgRNA plasmid library" could be applied.
4. In Fig3b, the knockdown efficiency of relative genes should be confirmed.
5. The authors used Sa-*evoBE4*(del-UGI) in HeLa to confirm the observation, how about Sp-*evoBE4*(del-UGI)? What is the rationale by choosing different Cas9?
6. Almost all base editing results in the main figure are presented using fold change, which magnifies the differences. Real efficiencies should be shown. Additionally, a differential statistical test should be provided when verifying in HEK293T and HeLa. Otherwise, it is impossible to judge whether this reduction has statistical significance.
7. Can the trans-factors be applied in base editor optimization?
8. Only two endogenous loci were selected for validation, which may not represent the general function of screened genes in gene editing. More loci should be included.
9. Some of the genes function differently in the K562, HeLa and HEK293T cell lines? Could the authors further explain and discuss?
10. In Fig4 and Fig5, not all the selected genes were verified. Is there any particular reason to exclude the two genes?

Minor points:

1. Three replicates should be shown in each bar, e.g. the bars in Fig5.
2. The full name should be used when first mentioned, e.g. NGS.
3. "Cas9n" and "nCas9" are confusingly used in the text.

Reviewer #2

(Remarks to the Author)

Title: Elucidating the genetic mechanisms governing cytosine base editing outcomes through CRISPRi screens

Summary: The authors developed two fluorescence reporters that allow cells that have undergone C•G to T•A or C•G to G•C cytosine base editing to be enriched and performed targeted CRISPRi screens in K562 cells to identify genetic determinants that affect these outcomes. From the screens, the authors found that C•G to T•A editing is promoted by MSH2 and MSH6 but suppressed by LIG3 and UNG (at least in some conditions), while C•G to G•C editing is promoted by RFWD3 and UNG but suppressed by XPF and XPA. The authors then performed arrayed experiments to validate hits at the reporter targets and two endogenous targets in K562 cells. Findings were also tested at endogenous targets in HeLa and 293T cells, although phenotypes from these experiments were somewhat inconsistent with those from K562s. In the end, the authors proposed a model of cytosine base editing.

Altogether, this manuscript represents a useful resource and contains many valuable insights. We find the MSH2 and MSH6 phenotypes interesting. We recommend addressing the following outstanding issues.

Comments (experimental):

1. Experiments were performed with cytosine base editors without UGI (SaCas9-BE4(UGI)/"SaCBE"). However, as mentioned in introduction (line 49-53), CBEs and CGBEs typically used for genome editing are those fused to UGI and UNG, respectively. We recommend that the authors confirm major hits (line 375: UNG, MSH2, MSH6, POLD1, LIG3, RFWD3, ERCC4, and XPA) with those editors. Such experiments may also help validate the model in Fig. 6, as inhibiting UNG by UGI should not affect the role of MSH2 and MSH6 in C•G to T•A editing.
2. The authors propose on line 75, line 525-536, and in the model that MSH2/MSH6 and LIG3 phenotypes were largely due to the nick on the G-containing strand. This hypothesis can be tested by asking if MSH2/6 and LIG3 knockdown affect editing when using CBEs that with dCas9 instead of Cas9 nickase (with or without UGI). If the hypothesis about the nick is correct, the phenotypes of MSH2/MSH6 and LIG3 should be dampened in the absence of the nick. As interpretation of the nick features prominently in the model, the mechanism should be directly evaluated.
3. As shown in Fig. 5 and discussed in line 470-473, the phenotypes of ERCC4, MSH2 and LIG3 in HeLa cells are the opposite compared to those in K562 cells. Because this manuscript focuses on mechanism, it is important that the phenotypes are consistent for majority of experiments OR that the inconsistencies can be reasonably explained. The authors suggest that such inconsistencies may be due to differences in editor delivery or editing kinetics. Not an unreasonable interpretation, but the authors should directly test the idea by evaluating the two editing strategies (stable/inducible verses transient overexpression) side-by-side in multiple cell lines using the same genetic perturbation strategy and same CBE in each. As of now, the comparisons have too many variables that could account for the inconsistencies.

To be clear, we think the inconsistencies observed for these phenotypes are interesting, especially given what has been reported previously, but the authors need to confirm the finding in head-to-head experiments and should test directly differences in cell lines, editor expression, even perhaps sequence/chromatin context of the targeted locus. The last of these possibilities seems plausible considering the effect in K562 cells is milder at HEK3 and RNF2 than the reporter.

Comments (text):

4. The manuscript is excessively long. We recognize that screens can provide a wealth of information, but we strongly recommend pairing the text down by (a) moving many technical details to the methods and (b) focusing results only on a few key phenotypes of high interest (perhaps those in Figure 6).
5. It's worth noting somewhere that 293T cells are mismatch repair deficient, which may change interpretation of phenotypes in those cells.
6. Please avoid interpretation of genes that did not score in the screen (as on line 530-531), while tempting to do so, there are many reasons, beyond lack of function in the process of interest, that could account for such results.

Reviewer #3

(Remarks to the Author)

Reviewer #4

(Remarks to the Author)

In the present manuscript titled "Elucidating the genetic mechanisms governing cytosine base editing outcomes through CRISPRi screens" the authors employed CRISPRi screens to identify endogenous DNA processing genes that influence C to T and C to G editing outcomes during cytosine base editing. Although there have been numerous reports on the impact of endogenous genes on cytosine base editing products, and some cellular genes have been identified to enhance C to G editing efficiency, there has not been a definitive report on the mechanisms underlying both editing outcomes. In this study, a

larger set of endogenous genes was screened using a fluorescence-based approach to improve the efficiency of gene selection. Ultimately, the authors conducted a comprehensive analysis of the relevant mechanisms, which holds significant scientific significance and provides valuable insights for optimizing cytosine base editing. Additionally, the authors could discuss why products may vary at different editing sites, even in host cells with the same genotype and using the same editor.

Version 1:

Reviewer comments:

Reviewer #1

(Remarks to the Author)

The authors have fully addressed my previous concerns.

Reviewer #2

(Remarks to the Author)

The authors have improved this manuscript. I only have the following recommendations prior to publication:

(1) Unless I missed something, the RNF2 and HEK3 sites were exclusively used after transduction. I would refrain from calling these endogenous sites, since they are used as exogenous loci. Rewording to avoid the term “endogenous” would be sufficient.

(2) I encourage the authors to mention that for each of their mechanistic conclusions alternative possibilities exist. Using language like “A possible explanation . . .” would be sufficient in most cases.

(3) The RFWD3 and “mixed editing outcomes” results sections could be condensed, in particular the rescue experiment in the RFWD3 section is really hard to interpret as adding back RFWD3 to the knockout line does not seem to rescue the observed effect. I worry that the discussion of this rescue errs on the side of over interpretation (i.e., if you cannot rescue the effect with wildtype RFWD3, then how do you know that the C315 residue is functional?). To be clear, without these two sections, I believe the manuscript is good enough. I’d just skip the mechanistic interpretation of that experiment and list the observations made instead, as in the “mixed editing outcomes” section.

Minor things, just to be helpful.

(4) I got a bit confused when the authors switched to discussing C-G to A-T, as opposed to C-G to T-A, at the end of the fifth results section. I figured it out, but it took a minute. Could be helpful to state more clearly that these are non-equivalent outcomes.

(5) In the sentence, “However, we did not observe knockdown of ERCC1 to significantly affect either the C-G to G-C or C-G to T-A outcome (Fig. 2F)”, I believe the wrong figure is called, as ERCC1 does not appear in Fig. 2F (although I did find the data in the paper).

Reviewer #4

(Remarks to the Author)

In this revised manuscript, the authors have addressed all the points raised by the reviewer. I have no further concerns, and it is suitable for publication in Nature Communications.

We would like to thank the reviewers for their thoughtful and constructive feedback regarding this work. We have made several major changes to the manuscript in response to this feedback (indicated in red in the manuscript and SI files), which we believe has greatly improved the rigor and quality of the manuscript. Below you will find a point-by-point response to each comment (in blue). Additionally, a summary of the major alternations/additions to the manuscript are:

1. We have pared down the first two results sections (which describe the development of the screening system and QC of the screening results) and moved much of the text to the Methods, figure captions, and supplementary discussion.
2. We have completed additional validation experiments in K562 cells using transient transfection of the base editor and gRNA, to better match the conditions in additional cell types such that we could do a proper head-to-head comparison. These new conditions resulted in bystander editing at both sites, but in a manner that enabled us to observe the impacts of knockdown on single C to T/G outcomes as well as dual C to T/G outcomes.
3. We expanded the base editors that we used to include rA1-SaBE4(Δ UGI) (which was the only construct tested previously), rA1-SaBE4 (which contains UGI and thus inhibits UNG in addition to the target gene that is knocked down), and rA1-SaBE1 (which utilizes dCas9 and thus does not introduce a nick on the strand opposite the uracil).
4. We performed validation experiments in HeLa and HEK293T knockdown cell lines under conditions (transient transfection of rA1-SaBE4(Δ UGI) and rA1-SaBE4) that match those of our K562 experiments such that we could robustly identify consistent trends across the three cell types.
5. These additional experiments revealed consistent contributions of MSH2/MSH6, UNG, and RFWD3 to processing of CBE intermediates.
6. Under these new conditions, the enhanced toxicity due to transfection caused high levels of cell death in the *POLD1* and *LIG3* knockdown cells, presumably due to the importance of these genes and their essential role in DNA repair and cell cycle regulation. Thus, we unfortunately did not observe significant changes in editing outcomes following knockdown of these genes when coupled with transfection of the base editor.
7. We have replaced all fold-change graphs with plots that show absolute editing efficiencies, with fold-change values displayed as heatmaps next to these graphs. Additionally, we changed the statistical tests used and now use adjusted z-scores in heatmaps to highlight significant changes.

Reviewer #1:

1. The authors spent too much space on the introduction of the CRISPRi system and quality control, which could be shown with bar graphs instead of long text. And most of the analysis process could be excluded from the result context, and move to the Method part.

We appreciate your suggestion. We have trimmed down the text in the “Coupling fluorescent reporters for CBE outcomes with CRISPRi screens” and “Identification of genes that are synthetic lethal to CBE expression and/or activity” sections and moved much of them to a Supplementary Discussion and the methods (these changes are highlighted).

2. The sequencing depth and real read counts in each editing outcomes should be included to better convince the readers.

This is a great consideration. We have added the real read counts and sequencing depth for each population in the new supplementary figure 4a-b as shown here.

For all libraries derived from different populations, we achieved a sequencing depth of at least 200X. Additionally, the number of zero-count sgRNAs and the Gini index for each library are presented in supplementary figures 4c-d. Zero-count sgRNAs account for at most 2.5% of the library, and the Gini indexes are generally below 0.1, indicating an even distribution of library members. Overall, these data demonstrate the high quality of our library preparation and sequencing metrics.

3. In the section "Identification of genes that are synthetic lethal to CBE expression and/or activity", the authors use "bulk, +dox samples" vs. "bulk, -dox samples" to screen out genes that are lethal to CBE function. Did they exclude genes that are essential for cell survival? To define essential genes, compare of “bulk, -dox samples” to “sgRNA plasmid library” could be applied.

Thanks for pointing this out. Both the “bulk, +dox samples” and “bulk, -dox samples” were first compared to the D0 sample (sgRNA library), which identified essential gene drop out (Supplementary Fig. 4f and Supplementary Fig. 17a). Therefore, the comparison between “bulk, +dox samples” and “bulk, -dox samples” should cancel out the dropout of essential genes and only reveal the genes that dropped out or enriched solely due to base editor

expression. In the revised manuscript, we have clarified this point and moved this section to the supplementary discussion to keep the manuscript more concise.

4. In Fig3b, the knockdown efficiency of relative genes should be confirmed.

Thank you very much for the suggestion. In the revised manuscript, Fig. 3b has been moved to Fig. 2d. We have confirmed knockdown efficiencies of key genes critical to our findings, including *UNG*, *MSH2*, *LIG3*, *RFWD3* and *ERCC4*, in K562, HeLa, and HEK293T cells, as shown in Supplementary Fig. 11, 14c, and 15b. We have also shown that the CRISPRi K562 cell line with dSpCas9-BFP-KRAB used in this experiment demonstrated strong knockdown capability exemplified by the significant dropout of essential gene during the screens (Supplementary Fig. 4f and Supplementary Fig. 17a). In addition, the CRISPRi sgRNAs used in this experiment are from the extensively validated hCRISPRi-v2 library, which was widely used in other studies.¹⁻³ While we agree that quantifying knockdown efficiencies in the relative cell lines (60 lines in total, or 26 lines that passed the z-score threshold) would be the most rigorous, this presents a significant technical challenge and does not substantially add value to the main findings of the paper. We will also note that in similar work, gene knockdown was not verified at this initial stage of hit validation.^{3,4}

5. The authors used Sa-evoBE4(Δ UGI) in HeLa to confirm the observation, how about Sp-evoBE4(Δ UGI)? What is the rationale by choosing different Cas9?

The main reason for using evo-SaBE4 (Δ UGI) in the HeLa cells is to avoid sgRNA crosstalk with the SpCas9-derived CRISPRi machinery used in these cells. The HEK293T knockdown cell lines were generated via transient transfection of the SpCas9-derived CRISPRoff vector. In these experiments, we waited 7 days for the CRISPRoff vector to degrade, which then allowed us to use evo-SpBE4 (Δ UGI) in these cell lines (Supplementary Fig. 15a and Supplementary Fig. 16a).

In the revised manuscript, to ensure consistency of base editor constructs when comparing across different expression systems and cell lines, we performed additional experiments by transfecting rA1-SaBE4(Δ UGI) and rA1-SaBE4 in K562, HeLa, and HEK293T knockdown cell lines (Supplementary Fig. 13a, Supplementary Fig. 14b and Supplementary Fig. 15c). In HEK293T cells, we also observed that the effect of *RFWD3* knockdown is consistent when editing with rA1-SaBE4(Δ UGI) and evo-BE4(Δ UGI).

6. Almost all base editing results in the main figure are presented using fold change, which magnifies the differences. Real efficiencies should be shown. Additionally, a differential statistical test should be provided when verifying in HEK293T and HeLa. Otherwise, it is impossible to judge whether this reduction has statistical significance.

Thank you very much for your suggestion. In the revised manuscript, we have replaced all fold change plots with plots that show absolute editing efficiencies (new Fig. 4-6). Fold changes are now displayed in heatmaps. Furthermore, we have introduced adjusted z-scores in heatmaps to highlight significant changes.

Specifically, we computed individual z-scores for each replicate:

$$z = \frac{x - \mu}{\sigma}$$

where μ is the mean of the negative controls, and σ is the standard deviation of the negative control.

Then we computed the adjusted z-score for multiple replicates:

$$\text{adjusted z-score} = |\text{mean}(z)| - \sigma(z)$$

where $|\text{mean}(z)|$ represents the absolute value of the average z-score from multiple replicates, and $\sigma(z)$ represents the standard deviation of the z-score distribution.

We annotated the heatmaps as follows:

*: $1.5 < \text{adjusted z-score} \leq 2$. **: $2 < \text{adjusted z-score} \leq 3$. ***: $\text{adjusted z-score} > 3$.

Essentially, * indicates that the distribution of editing frequencies from a knockdown cell line lies primarily between 93.3% to 97.7% away from the negative control distribution. ** indicates that the distribution of editing frequencies from a knockdown cell line lies primarily between 97.7% to 99.9% away from the negative control distribution. *** indicates that the distribution of editing frequencies from a knockdown cell line lies primarily 99.9% away from the negative control distribution.

This has been clarified in the Methods section of the manuscript as well.

7. Can the trans-factors be applied in base editor optimization?

Thank you very much for the suggestion. Previous studies have spent extensive efforts to optimize base editors through rational designs based on DNA repair mechanisms.^{3,5}

We explored several strategies to enhance C•G to G•C editing outcomes, by either fusing certain trans-factors to the C-terminus of evoA1-Cas9n or over-expressing them from a separate plasmid. These plasmids were transfected into HEK293T cells incorporated with a C•G to G•C fluorescent reporter, and turn-on rates were quantified three days post transfection using flow cytometry (shown below). Unfortunately, none of these strategies increased the C•G to G•C fluorescent turn-on rate compared to the evoA1-Cas9n baseline and thus we have elected to not include them in the manuscript.

Explanation of constructs:

1. evoA1-Cas9n-RPA2: C-terminus fusion of RPA2. RPA2 (Replication Protein A2) binds ssDNA and interacts with many DNA repair factors.⁶ Specifically, the winged helix domain (WHD) of RPA2 interacts with several validated factors identified in the screens, including UNG, RFW3 and ERCC4.⁷
2. evoA1-Cas9n-WHD: C-terminus fusion of WHD.
3. evoA1-Cas9n-RPA2(del238-269): C-terminus fusion of RPA2 with truncation from amino acid 238 to 269. Amino acid 238-269 is the specific region in WHD that interacts with UNG and RFW3. This serves as a control construct for RPA2 fusion.
4. evoA1-Cas9n-RPA2(delWHD): C-terminus fusion of RPA2 with truncation of WHD. This serves as a control construct for RPA2 fusion.
5. evoA1-Cas9n-WHD(del238-269): C-terminus fusion of WHD with truncation from amino acid 238 to 269. This serves as a control construct for WHD fusion.

6. evoA1-Cas9n-hUNG(FL): C-terminus fusion of full length human UNG2.
7. evoA1-Cas9n-hUNG(del1-68): C-terminus fusion of human UNG2 with truncation from amino acid 1 to 68. The N-terminus 1-68 of hUNG2 interacts with PCNA and WHD of RPA2.⁸ This serves as a control construct for hUNG2(FL) fusion.
8. evoA1-Cas9n-RFWD3: C-terminus fusion of RFWD3.
9. evoA1-Cas9n-ERCC4(del2-639, D715A): C-terminus fusion of a potential dominant negative form of ERCC4. Deletion of amino acid 2 to 639 removes the helicase domain of ERCC4, while preserving the nuclease catalytic domain and ERCC1-interacting domain.⁹ The D715A mutation deactivates nuclease activity.
10. evoA1-Cas9n-ERCC4(del2-639, K727A): C-terminus fusion of a potential dominant negative form of ERCC4. The K727A mutation deactivates nuclease activity.
11. evoA1-Cas9n + ERCC4(D715A). Overexpression of a deactivated ERCC4.
12. evoA1-Cas9n + ERCC4(K727A). Overexpression of a deactivated ERCC4.
13. evoA1-Cas9n + ERCC4(del2-639, D715A). Overexpression of a potential dominant negative form of ERCC4.
14. evoA1-Cas9n + ERCC4(del2-639, K727A). Overexpression of a potential dominant negative form of ERCC4.

Because of the negative outcome of these experiments, we elected to omit them from the manuscript and instead focus on experiments that focus on understanding the mechanism of cytosine base editing. In the revised manuscript, we performed experiments where we over-expressed RFWD3 (to increase C•G to G•C editing efficiencies), as well as two mutants with impaired function (to decrease the C•G to G•C outcome) as we believe these experiments provide information on the mechanism of RFWD3-mediated processing of the CBE intermediate. While all overexpression experiments primarily led to a reduction in absolute editing efficiencies (Fig. 5h), potentially due to resource competition between

expression of the CBE and the large RFW3 gene, changes in the relative rates of C•G to G•C and C•G to T•A outcomes were analyzed and included.

8. Only two endogenous loci were selected for validation, which may not represent the general function of screened genes in gene editing. More loci should be included.

Thank you very much for the suggestion. To evaluate for the generality of genes identified from the screens, we have tested the impact of gene knockdown at two genomically integrated “synthetic” sites (the BFP and dGFP reporters) in K562 cells, as well as the two endogenous sites in K562, HeLa, and HEK293T cells. This is consistent with other work, in which only a single endogenous site was used for validation, or Repair-seq-type screens, in which hits are validated at a variety of sequences, but all within a genomically-integrated, “synthetic” site.^{2,10} In new experiments in this manuscript, we first expanded our testing of base editor variants at these two endogenous loci across three different cell lines, and with multiple editors. Specifically, we tested rA1-SaBE4(Δ UGI), rA1-SaBE4, and rA1-SaBE1 in K562 knockdown cell lines via transient transfection (Fig. 6a-b). Additionally, we tested rA1-SaBE4(Δ UGI) and rA1-SaBE4 in HeLa and HEK293T knockdown cell lines via transient transfection. As the bystander Cs were also edited in these transfection systems, we were able to analyze the impact of gene knockdown on a total of 4 different target Cs. These experiments revealed:

- 1) The gene knockdown effects varied depending on different editing kinetics. For example, editing efficiencies were significantly higher in K562 cells transfected with rA1-SaBE4(Δ UGI) compared to K562 cells with integrated rA1-SaBE4(Δ UGI) (Fig. 3a-b and Supplementary Fig. 13b-c). In general, knockdown of *MSH2* and *MSH6* caused a less substantial reduction in C•G to T•A editing outcomes in the transfected system than in the integrated system (Fig. 4a and 4d), and required concurrent inhibition of UNG (with the rA1-SaBE4 construct) to observe impacts in gene knockdown. Notably, the integrated system mainly produced editing outcomes in which a single C within the protospacer was edited (presumably derived from a single uracil intermediate, which matches the system on which we ran our screen), whereas dual editing outcomes are more predominant in the transfection system (presumably derived from dual uracil intermediates). Due to the wide editing window of SaCas9-derived CBEs, finding protospacers with only one target C was particularly challenging.
- 2) The gene knockdown effects varied depending on the target base positions. While we have shown the effects of *MSH2* and *MSH6* knockdown become more prominent when editing with rA1-SaBE4, which co-inhibits UNG, we observed that their influence depends on the target C position. At *HEK3* across all three cell lines, knockdown of *MSH2* reduced the single C•G to T•A editing frequencies at the C10 position but increased the single C•G to T•A editing frequencies at the C16 position (Fig. 4a-c). Since the C16 position is only one base pair away from the Cas9-induced nick, it may be processed productively (to produce a C•G to T•A outcome) via a pathway independent of MutS α recognition.
- 3) The gene knockdown effects due to *ERCC4* varied depending on different cell types. We observed discrepancies in the knockdown effects of *ERCC4* between K562 and HeLa cells (Fig. 6a-b and Supplementary Fig. 17). While *ERCC4* knockdown leads to an increase in base editing efficiencies across nearly all outcomes in K562 and HEK293T cells, we observed a minimal effect on editing outcomes in HeLa cells.

These head-to-head comparisons at the *HEK3* and *RNF2* sites across these three cell lines uncovered that *MSH2* knockdown consistently decreased certain C•G to T•A editing outcomes when *UNG* was inhibited, and *RFWD3* knockdown consistently decreased certain C•G to G•C editing outcomes. We further validated the effect of *RFWD3* knockdown in 5 additional sites in HEK293T cells (Supplementary Fig. 15f-g).

9. Some of the genes function differently in the K562, HeLa and HEK293T cell lines? Could the authors further explain and discuss?

In the previous version of this manuscript, we compared gene knockdown effects between an integrated rA1-SaBE4(Δ UGI) system in K562 cells, and a transiently transfected evo-SaBE4(Δ UGI) (HeLa cells) or evo-SpBE4(Δ UGI) (HEK293T cells). Because of the drastic differences of these various systems, we performed additional experiments in the revised manuscript in which the conditions across the different cell lines were held more constant. Specifically, to first assess the impact of CBE delivery (which impacts editing kinetics), we compared gene knockdown effects in K562 cells transfected with rA1-SaBE4(Δ UGI) to those in K562 cells with the integrated rA1-SaBE4 (Δ UGI) (Fig. 3a-b and Fig. 6a-b). Then, to evaluate cell type differences, we compared gene knockdown effects across K562, HeLa, and HEK293T cells transfected with rA1-SaBE4(Δ UGI) and rA1-SaBE4.

These additional experiments revealed that the increase in editing kinetics due to the transient transfection of the CBE resulted in a widened editing window and thus more than one C getting edited. In a side-by-side comparison, the integrated CBE system primarily generated single C to T editing products, whereas the transfection system frequently generated dual C to T editing products, which are likely produced by different intermediates (single uracil versus dual uracil). In these new experiments, we have analyzed these different outcomes (single C to T, dual C to T) and found that *RFWD3* and *MSH2* knockdown are consistent across the three cell types.

10. In Fig4 and Fig5, not all the selected genes were verified. Is there any particular reason to exclude the two genes?

Thank you for your detailed review of the manuscript. In K562 cells, we characterized all 7 genes, *UNG*, *RFWD3*, *MSH2*, *MSH6*, *POLD1*, *ERCC4* and *LIG3*. In HeLa cells, we omitted *MSH6* because it forms a complex with *MSH2* and exhibits very similar knockdown effects. Therefore, we selected *MSH2* to represent the entire MutS α complex.

We also omitted *POLD1* because its knockdown is known to affect cell cycle progression.¹¹ Transfection caused severe toxicity in the *POLD1* (and *LIG3*) knockdown K562 experiments (data not shown). We believe it may contribute to the discrepancies observed in *POLD1* and *LIG3* knockdown effects between the transfected and the integrated systems. Therefore, we decided not to further pursue *POLD1* in HeLa cells.

In the revised manuscript, we decided to focus on the just *MSH2* and *RFWD3* in HEK293T cells, whose knockdown effects are consistent across K562 and HeLa cells.

Minor points:

1. Three replicates should be shown in each bar, e.g. the bars in Fig5.

Thank you very much for your suggestions. In the revised manuscript, we have replaced the old Fig. 5 with new Fig. 4-6, which now display individual data points.

2. The full name should be used when first mentioned, e.g. NGS.

Thank you for your attention to details. This has been fixed.

3. "Cas9n" and "nCas9" are confusingly used in the text.

Thank you for catching that. We have made sure to use "Cas9n" consistently throughout the revised manuscript.

Reviewer #2:

Comments (experimental):

1. Experiments were performed with cytosine base editors without UGI (SaCas9-BE4(DUGI)/"SaCBE"). However, as mentioned in introduction (line 49-53), CBEs and CGBEs typically used for genome editing are those fused to UGI and UNG, respectively. We recommend that the authors confirm major hits (line 375: UNG, MSH2, MSH6, POLD1, LIG3, RFWD3, ERCC4, and XPA) with those editors. Such experiments may also help validate the model in Fig. 6, as inhibiting UNG by UGI should not affect the role of MSH2 and MSH6 in C•G to T•A editing.

Thank you for your thorough review of our manuscript and this suggestion. In the revised manuscript, we performed additional experiments by transfecting K562, HeLa and HEK293T knockdown cell lines with both rA1-SaBE4 (Δ UGI) and rA1-SaBE4, which contains 2 copies of UGI. We used rA1-SaBE4 (Δ UGI) as the "CGBE" because it demonstrated comparable C•G to G•C purities to those achieved with UNG fusion.³ Also, the SpCas9 counterpart is regarded as a "miniCGBE".^{3,12} This experiment revealed several key findings in the revised manuscript.

- 1) While *MSH2* and *MSH6* knockdown did not cause as substantial a reduction of C•G to T•A editing frequencies in the transfection system in K562, HeLa, or HEK293T cells with rA1-SaBE4 (Δ UGI) compared to the integrated system in K562 cells (Fig. 3a-b and Fig. 4), consistent with previous studies^{3,5}, we did observe pronounced and consistent changes in editing outcomes across all three cell lines when UNG was simultaneously inhibited via the use of the rA1-SaBE4 construct (Fig. 4). Notably, transfection led to much higher overall editing efficiencies within a shorter time frame, along with editing of bystander Cs (Fig. 3a-b, and Supplementary Fig13. b-c). These changes in the editing profile may be responsible for the dominance of UNG in processing the intermediates under these conditions.
- 2) Specifically, knockdown of either *MSH2* or *MSH6* led to significant decreases in the frequencies of the single C•G to T•A editing outcome at the C10 position at both the *HEK3* and *RNF2* sites when using rA1-SaBE4. Because Mut α and UNG both recognize the U•G mismatch, these observations suggest that Mut α and UNG compete for substrate recognition, with UNG having a competitive advantage.
- 3) Intriguingly, we also found the effect of Mut α knockdown to be dependent on the target base position. At the *HEK3* site, knockdown of *MSH2* or *MSH6* had minimal impact on the dual C•G to T•A outcome frequency but led to a significant *increase* in the frequency of the single C•G to T•A editing outcome at the C16 position (Fig. 4a). Since the C16 position is only one base pair away from the Cas9-induced nick, it may be processed productively (to produce a C•G to T•A outcome) via a pathway independent of Mut α recognition. When the target C is shifted two base pairs away from the nick to the C14 position, as in *RNF2*, knockdown of *MSH2* or *MSH6* did not result in a significant increase in the single C•G to T•A editing frequency at this position (Fig. 4d).
- 4) The effects of *MSH2* knockdown on editing with rA1-SaBE4 were consistent in HeLa and HEK293T cells. As you point out in a later comment, HEK293T cells are MMR deficient. This deficiency is attributed to the silencing of the MutL α complex, while the expression levels of Mut α remain normal. The comparable editing efficiencies between HEK293T and HeLa cells (Supplementary Fig. 14d-e and Supplementary Fig. 15d-e) combined with the

similar impacts of *MSH2* knockdown on editing efficiencies across these cell lines demonstrate that mismatch recognition is required, but downstream nicking by MutL α is dispensable for cytosine base editing when using Cas9n-derived base editors.

- 5) Editing with rA1-SaBE4 predominately produces C•G to T•A editing outcomes. Therefore, genes such as *RFWD3* and *ERCC4*, which mainly influence C•G to non-T•A editing or mixed editing outcomes, did not exhibit a major impact on editing with rA1-SaBE4.
- 6) We tested rA1-SaBE1 (dCas9 instead of Cas9n) in K562 knockdown cells as well. The editing efficiencies by this construct in the NC cell lines were quite low (less than 5%) and thus observing statistically significant further reductions in editing due to gene knockdown would be difficult (Fig. 4a and d). Nevertheless, we found that *MSH6* knockdown caused a reduction in the single C•G to T•A editing outcome at the C10 position at the *RNF2* site, from $1.9 \pm 0.8\%$ to $0.8 \pm 0.1\%$ (representing a 2.5 ± 0.8 -fold decrease, Fig. 4d).

2. The authors propose on line 75, line 525-536, and in the model that MSH2/MSH6 and LIG3 phenotypes were largely due to the nick on the G-containing strand. This hypothesis can be tested by asking if MSH2/6 and LIG3 knockdown affect editing when using CBEs that with dCas9 instead of Cas9 nickase (with or without UGI). If the hypothesis about the nick is correct, the phenotypes of MSH2/MSH6 and LIG3 should be dampened in the absence of the nick. As interpretation of the nick features prominently in the model, the mechanism should be directly evaluated.

Thank you for this suggestion. We have conducted additional experiments by transfecting our K562 knockdown cell lines with rA1-SaBE4(Δ UGI), rA1-SaBE4 and rA1-SaBE1 (Fig. 6a-b) to perform a direct head-to-head comparison. We did not observe statistically significant changes in editing upon *LIG3*, *MSH2/6*, or *POLD1* knockdown in the rA1-SaBE1 experiments (consistent with our hypothesis), but the baseline editing was quite low (less than 5%) with this editor, making it difficult to observe further decreases in editing due to knockdown (knockdown of *UNG* significantly increased C to T editing with this editor though).

Further, we did not observe a significant increase in C•G to T•A editing outcomes with *LIG3* knockdown in cells transfected with rA1-SaBE4(Δ UGI), which contrasts with our observations in the integrated system (Fig. 3a-b). Changes in editing kinetics and thus editing intermediates (single vs dual uracils) could potentially contribute to these discrepancies. Furthermore, we observed that transfection causes enhanced toxicity in *LIG3* (and *POLD1*) knockdown cells compared to control and other knockdown cell lines (this was consistent across all cell lines). This may have caused transfected, *LIG3* knockdown cells to drop out of the cell population. While generating an integrated rA1-SaBE1 K562 cell line for a head-to-head comparison is possible, we are concerned that editing efficiencies in such a cell line would be too low to draw robust conclusions, as editing efficiencies from transfecting rA1-SaBE1 are already very low. Therefore, we have revised and weakened our statement regarding *LIG3*.

3. As shown in Fig. 5 and discussed in line 470-473, the phenotypes of *ERCC4*, *MSH2* and *LIG3* in HeLa cells are the opposite compared to those in K562 cells. Because this manuscript focuses on mechanism, it is important that the phenotypes are consistent for majority of experiments OR that the inconsistencies can be reasonably explained. The

authors suggest that such inconsistencies may be due to differences in editor delivery or editing kinetics. Not an unreasonable interpretation, but the authors should directly test the idea by evaluating the two editing strategies (stable/inducible versus transient overexpression) side-by-side in multiple cell lines using the same genetic perturbation strategy and same CBE in each. As of now, the comparisons have too many variables that could account for the inconsistencies.

To be clear, we think the inconsistencies observed for these phenotypes are interesting, especially given what has been reported previously, but the authors need to confirm the finding in head-to-head experiments and should test directly differences in cell lines, editor expression, even perhaps sequence/chromatin context of the targeted locus. The last of these possibilities seems plausible considering the effect in K562 cells is milder at HEK3 and RNF2 than the reporter.

We really appreciate your suggestion. We agree that there were too many variables when comparing across different cell lines in the previous manuscript. Therefore, we performed additional experiments in the revised manuscript in which the conditions across the different cell lines were held more constant. Specifically, to first assess the impact of CBE delivery (which can impact editing rates), we compared gene knockdown effects in K562 cells transfected with rA1-SaBE4(Δ UGI) to those in K562 cells with the integrated rA1-SaBE4(Δ UGI) (Fig. 3a-b and Fig. 6a-b). Then, to evaluate cell type differences, we compared gene knockdown effects across K562, HeLa, and HEK293T cells transfected with rA1-SaBE4(Δ UGI) and rA1-SaBE4.

Head-to-head comparisons between the transient transfection system and stable inducible system in K562 cells revealed significant differences in knockdown effects (Fig. 3a-b and Fig. 6a-b). We have summarized the major findings in our previous responses. Notably, our observations align with published studies showing that *MSH2* and *MSH6* knockdowns do not significantly affect editing via transfection of rA1-SaBE4(Δ UGI). Interestingly, the effects of *MSH2* or *MSH6* knockdown were pronounced when UNG was simultaneously inhibited via the use of the rA1-SaBE4 construct. Two potential factors may synergize to give UNG such a competitive advantage over MutS α in the transfection system. First, multi-uracil intermediates are not optimal substrates for MutS α .¹³ In addition, multiple uracils could enhance the recruitment of UNG to the editing site.

Sequence and chromatin context can definitely affect the cellular processing of base editing intermediates as well. In addition to observing milder knockdown effects at the two endogenous sites compared to lentiviral reporter sites, we also noted different knockdown effects between *HEK3* and *RNF2*. Findings from our additional experiments underscore the complexity of base editing mechanisms, which vary depending on different editors, target loci/positions, and editing kinetics. However, we also note consistent mechanisms. *MSH2* knockdown consistently decreased certain C•G to T•A editing outcomes when *UNG* was inhibited, and *RFWD3* knockdown consistently decrease certain C•G to G•C editing outcomes.

Comments (text):

4. The manuscript is excessively long. We recognize that screens can provide a wealth of information, but we strongly recommend pairing the text down by (a) moving many technical details to the methods and (b) focusing results only on a few key phenotypes of high interest (perhaps those in Figure 6).

Thank you for your suggestion. We have trimmed down the text in the “Coupling fluorescent reporters for CBE outcomes with CRISPRi screens” and “Identification of genes that are synthetic lethal to CBE expression and/or activity” sections and moved much of them to a Supplementary Discussion and the methods (these changes are highlighted).

5. It’s worth noting somewhere that 293T cells are mismatch repair deficient, which may change interpretation of phenotypes in those cells.

Thank you for pointing this out. In the revised manuscript, we mention that HEK293T cells are MMR deficient due to silencing of the MutL α complex in the section “MutS α and UNG compete to process cytosine base editing intermediates” (text highlighted). It is very intriguing that we observed *MSH2* knockdown still decrease certain C•G to T•A editing outcomes when editing with rA1-SaBE4 in HEK293T cells. We therefore think mismatch recognition is still required, but downstream nicking by MutL α is dispensable for cytosine base editing when using Cas9n-derived editors.

6. Please avoid interpretation of genes that did not score in the screen (as on line 530-531), while tempting to do so, there are many reasons, beyond lack of function in the process of interest, that could account for such results.

Thank you for your suggestion. We have moved line 358-371 in the previous manuscript, which discussed genes hypothesized to participate in base editing but did not score in the screen, to the supplementary discussion. Due to the data from our new experiments in HEK293T cells, the discussion regarding MutS α has now been moved to the results section “MutS α and UNG compete to process cytosine base editing intermediates”.

Reviewer #3:

Thank you for taking the time to review this manuscript. We greatly appreciate your constructive feedback and have performed experiments to address your comments.

Reviewer #4:

In the present manuscript titled "Elucidating the genetic mechanisms governing cytosine base editing outcomes through CRISPRi screens" the authors employed CRISPRi screens to identify endogenous DNA processing genes that influence C to T and C to G editing outcomes during cytosine base editing. Although there have been numerous reports on the impact of endogenous genes on cytosine base editing products, and some cellular genes have been identified to enhance C to G editing efficiency, there has not been a definitive report on the mechanisms underlying both editing outcomes. In this study, a larger set of endogenous genes was screened using a fluorescence-based approach to improve the efficiency of gene selection. Ultimately, the authors conducted a comprehensive analysis of the relevant mechanisms, which holds significant scientific significance and provides valuable insights for optimizing cytosine base editing. Additionally, the authors could discuss why products may vary at different editing sites, even in host cells with the same genotype and using the same editor.

Thank you so much for reviewing the manuscript. We appreciate your positive feedback. You raised a very interesting question. For example, the distribution of editing outcomes varies largely between *HEK3* and *RNF2* sites in K562 cells, regardless of whether rA1-SaBE4(Δ UGI) is integrated (Fig. 3a-b) or transfected (Fig. 6a-b). Many genes influence base editing outcomes, and impacts of their knockdown differ between these two sites as evidenced by the heatmaps in Fig. 6a-b. We believe variations in editing outcomes can be partially attributed to differences in the distribution of pathways involved in processing base editing intermediates at different sites. Numerous factors, including target sequences, the number and positions of target Cs, and local chromatin context, likely contribute to the different distributions of processing pathways. In addition, target sequence and chromatin context also influence editing kinetics, which can eventually affect distribution of editing outcomes. We have incorporated these discussions in both the result and discussion sections.

References:

1. Horlbeck, M. A. *et al.* Compact and highly active next-generation libraries for CRISPR-mediated gene repression and activation. *eLife* **5**, e19760 (2016).
2. Hussmann, J. A. *et al.* Mapping the genetic landscape of DNA double-strand break repair. *Cell* **184**, 5653-5669.e25 (2021).
3. Koblan, L. W. *et al.* Efficient C•G-to-G•C base editors developed using CRISPRi screens, target-library analysis, and machine learning. *Nat Biotechnol* **39**, 1414–1425 (2021).
4. Chen, P. J. *et al.* Enhanced prime editing systems by manipulating cellular determinants of editing outcomes. *Cell* **184**, 5635-5652.e29 (2021).
5. Huang, M. E. *et al.* C-to-G editing generates double-strand breaks causing deletion, transversion and translocation. *Nat Cell Biol* **26**, 294–304 (2024).
6. Dueva, R. & Iliakis, G. Replication protein A: a multifunctional protein with roles in DNA replication, repair and beyond. *NAR Cancer* **2**, zcaa022 (2020).
7. Maréchal, A. & Zou, L. RPA-coated single-stranded DNA as a platform for post-translational modifications in the DNA damage response. *Cell Res* **25**, 9–23 (2015).
8. Kavli, B. *et al.* RPA2 winged-helix domain facilitates UNG-mediated removal of uracil from ssDNA; implications for repair of mutagenic uracil at the replication fork. *Nucleic Acids Research* **49**, 3948–3966 (2021).
9. Jones, M. *et al.* Cryo-EM structures of the XPF-ERCC1 endonuclease reveal how DNA-junction engagement disrupts an auto-inhibited conformation. *Nat Commun* **11**, 1120 (2020).
10. Richardson, C. D. *et al.* CRISPR–Cas9 genome editing in human cells occurs via the Fanconi anemia pathway. *Nat Genet* **50**, 1132–1139 (2018).
11. Tumini, E., Barroso, S., -Calero, C. P. & Aguilera, A. Roles of human POLD1 and POLD3 in genome stability. *Sci Rep* **6**, 38873 (2016).

12. Kurt, I. C. *et al.* CRISPR C-to-G base editors for inducing targeted DNA transversions in human cells. *Nat Biotechnol* **39**, 41–46 (2021).
13. Edelbrock, M. A., Kaliyaperumal, S. & Williams, K. J. Structural, molecular and cellular functions of MSH2 and MSH6 during DNA mismatch repair, damage signaling and other noncanonical activities. *Mutation Research/Fundamental and Molecular Mechanisms of Mutagenesis* **743–744**, 53–66 (2013).

Reviewer #1 (Remarks to the Author):

The authors have fully addressed my previous concerns.

Reviewer #2 (Remarks to the Author):

The authors have improved this manuscript. I only have the following recommendations prior to publication:

(1) Unless I missed something, the RNF2 and HEK3 sites were exclusively used after transduction. I would refrain from calling these endogenous sites, since they are used as exogenous loci. Rewording to avoid the term “endogenous” would be sufficient.

In these experiments, the sgRNAs targeting these sites were transduced, but the sites are endogenous. We have changed the wording in the manuscript to make this clear.

(2) I encourage the authors to mention that for each of their mechanistic conclusions alternative possibilities exist. Using language like “A possible explanation . . .” would be sufficient in most cases.

We have tempered our conclusions (throughout the results sections as well as the discussion) by using more compliant language (indicated in red text).

(3) The RFWD3 and “mixed editing outcomes” results sections could be condensed, in particular the rescue experiment in the RFWD3 section is really hard to interpret as adding back RFWD3 to the knockout line does not seem to rescue the observed effect. I worry that the discussion of this rescue errs on the side of over interpretation (i.e., if you cannot rescue the effect with wildtype RFWD3, then how do you know that the C315 residue is functional?). To be clear, without these two sections, I believe the manuscript is good enough. I'd just skip the mechanistic interpretation of that experiment and list the observations made instead, as in the “mixed editing outcomes” section.

This section had been edited accordingly (indicated in red text)

Minor things, just to be helpful.

(4) I got a bit confused when the authors switched to discussing C-G to A-T, as opposed to C-G to T-A, at the end of the fifth results section. I figured it out, but it took a minute. Could be helpful to state more clearly that these are non-equivalent outcomes.

We have added an additional sentence at the beginning of this paragraph to clarify this.

(5) In the sentence, “However, we did not observe knockdown of ERCC1 to significantly affect either the C-G to G-C or C-G to T-A outcome (Fig. 2F)”, I believe the wrong figure is called, as ERCC1 does not appear in Fig. 2F (although I did find the data in the paper).

This has been fixed (the data are in Supplementary Figure 8)

Reviewer #4 (Remarks to the Author):

In this revised manuscript, the authors have addressed all the points raised by the reviewer. I have no further concerns, and it is suitable for publication in Nature Communications.